

# Numerical Case Study of the Aerosol-Cloud-Interactions in Warm Boundary Layer Clouds over the Eastern North Atlantic with an Interactive Chemistry Module

Hsiang-He Lee[1], Xue Zheng[1], Shaoyue Qiu[1], and Yuan Wang[2]

[1]Atmospheric, Earth, and Energy Division, Lawrence Livermore National Laboratory, Livermore, CA, U.S.A.
[2]Department of Earth System Science, Stanford University, Stanford, CA, U.S.A.


*Corresponding author address: Dr. Hsiang-He Lee, 7000 East Avenue, Livermore, CA, 94550, U.S.A.

E-mail: lee1061@llnl.gov



## Abstract

The presence of warm boundary layer stratiform clouds over the Eastern North Atlantic (ENA) region is commonly influenced by the Azores High, especially during the summer season. To investigate comprehensive aerosol-cloud interactions, this study employs the Weather Research and Forecast model coupled with a chemistry component (WRF-Chem), incorporating aerosol chemical components that are relevant with formation of cloud condensation nuclei (CCN) and accounting for aerosol spatiotemporal variation. This study focuses on aerosol indirect effects, particularly long-range transport aerosols, in the ENA region under three different weather regimes: ridge with surface high-pressure system, post-trough with surface high-pressure system, and weak trough. The WRF-Chem simulations conducted at a near the Large-Eddy Simulation scale offer valuable insights into the model's performance, especially regarding its high spatial resolution in accurately capturing the liquid water path (LWP) and cloud fraction across various weather regimes. Our result shows that introducing five times more aerosols to either non-precipitating or precipitating clouds significantly increases ambient CCN numbers, resulting in varying degrees of higher LWP. The substantial aerosol-cloud interaction especially occurs in the precipitating clouds and demonstrates the LWP susceptibility to changes in CCN under different regimes. Conversely, non-rain clouds at the edges of a cloud system are prone to evaporation, exhibiting an aerosol drying effect. The aerosols released during this process transition back to the accumulation mode, facilitating future activation. This dynamic behavior is not adequately represented in prescribed-aerosol simulations.



# 1 Introduction


Low-level stratiform clouds are predominantly generated over oceanic regions and are
categorized into three main types: warm boundary layer stratiform clouds located on the eastern
side of oceanic subtropical highs, stratocumulus clouds that develop over warm western boundary
currents during winter cold outbreaks, and Arctic stratus (Klein and Hartmann, 1993). Warm
boundary layer stratocumulus clouds, on average, blanket around 20% of the Earth's surface
annually (Wood, 2012; Warren et al., 1988). Their influence on the Earth's energy balance is
substantial, primarily through their ability to reflect incoming solar radiation, resulting in
significant shortwave cloud radiative effects leading to a pronounced negative net radiative effect
(Chen et al., 2000; Stephens and Greenwald, 1991; Hartmann et al., 1992).
Research on aerosol-cloud interactions in warm boundary layer clouds has been ongoing
since the 1970s. Twomey (1974) proposed that aerosols play an important role in influencing the
Earth's energy budget by serving as cloud condensation nuclei (CCN). These CCN are crucial for
cloud formation. A higher concentration of CCN results in the formation of clouds with a greater
number of smaller-sized cloud droplets (Twomey, 1991). These smaller droplets enhance cloud
albedo, known as the first indirect effect, and inhibit precipitation formation while prolonging
cloud lifetime, known as the second indirect effect (Albrecht, 1989). In addition to these indirect
effects, aerosol particles have direct, semi-direct, and indirect impacts on the atmosphere's energy
budgets and surface, leading to changes in atmospheric stability (Lee et al., 2008). Until now, our
understanding of aerosol-cloud interactions remains incomplete. In a recent review paper,
Feingold et al. (2024) highlighted that the response of cloud amount (including liquid water
content, spatial coverage, and cloud persistence) to aerosol perturbations is still unclear. Both
positive and negative adjustments in liquid water path (LWP) and cloud fraction (CF) have been

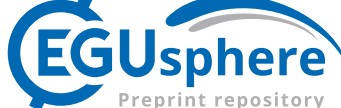

observed. Increases in cloud amount (positive adjustments) are linked to rain suppression, whereas
enhanced evaporation of smaller droplets and entrainment feedback tend to decrease cloud amount
(negative adjustments).

This study focuses on warm boundary layer stratiform clouds located on the eastern side

of oceanic subtropical highs, specifically targeting the area over the Eastern North Atlantic (ENA)
region, where the U.S. Department of Energy (DOE) Atmospheric Radiation Measurement (ARM)
program developed a ground-based user facility in the Azores archipelago (Mather and Voyles,
2013). Long-term ground-based observations at the ARM ENA site, aircraft field campaigns near
the Azores islands, and satellite retrievals over the ENA region provide comprehensive datasets
for observational studies on aerosol-cloud interactions (Zheng et al., 2022; Zheng et al., 2023;
Ghate et al., 2023; Qiu et al., 2024).

The presence of stratocumulus clouds over the ENA region is commonly influenced by the

Azores High, also known as the Bermuda-Azores High (Rémillard and Tselioudis, 2015). This
semi-permanent high-pressure system typically develops over the subtropical North Atlantic
Ocean. The Azores High often brings stable and relatively dry conditions to the region, which can
contribute to the formation and maintenance of stratocumulus clouds. During the summer season,
the Bermuda-Azores High tends to strengthen and expand, leading to more persistent high-pressure
conditions and often warmer, drier weather in its vicinity. Although synoptic intrusions from high
latitudes are less frequent in the summer compared to the winter season (Wood et al., 2015), the
ENA region still experiences synoptic variability from weak troughs during the summer months
(Mechem et al., 2018; Zheng et al., 2024).



Leveraging the marine boundary layer cloud observations from the ARM ENA
observatory, this study aims to study aerosol indirect effects (AIE), especially long-range transport
aerosols, in the warm boundary layer clouds over the ENA region under three different synoptic
regimes: ridge with surface high-pressure system, post-trough with surface high-pressure system,
and weak trough (Mechem et al., 2018; Zheng et al., 2024).  These regimes are chosen because,
during them, the ARM site experiences northerly wind conditions, minimizing the influence of the
island effect on the observations (Ghate and Cadeddu, 2019; Zheng and Miller, 2022).
Only a few numerical studies examined aerosol-cloud interactions in marine boundary
layer clouds over this region (Zhang et al., 2021; Wang et al., 2020; Kazemirad and Miller, 2020;
Christensen et al., 2024).  Wang et al. (2020), for example, used the Weather Research and Forecast
(WRF) model with prescribed CCN profiles to simulate perturbed long-range transport aerosol
concentration for two different cases of marine boundary layer (MBL) clouds.  They concluded
that when long-range transport aerosol plumes penetrate down into the drizzling cloud deck, the
simulations show an increase in marine cloud fractions with larger water content, supporting a
positive cloud amount adjustment to CCN perturbations.  Christensen et al. (2024) utilized an
advanced WRF configuration integrated with a Lagrangian framework to assess the effects of
aerosols on developing cloud fields across 10 case study days during the ENA field campaign and
got the same conclusion.  However, a limitation of these studies is that they do not account for
aerosol composition acting as CCN or the changes in aerosol populations following the cloud
evaporation process, even though aerosol wet removal is included in their simulations.
To further investigate the impacts of realistic aerosol chemical components and aerosol
spatiotemporal variation on the AIE, this study adopts the WRF model coupled with a chemistry
component (WRF-Chem) to examine the AIE in the ENA region across different synoptic regimes.



A brief description of observational data and the WRF-Chem model, as well as the configuration
and numerical experiments, are given in Section 2.  Simulated results are discussed in Section 3,
including model evaluation, model sensitivity tests, and cloud susceptibilities.  The discussion and
summary are provided in Section 4.

# 2   Methodology

## 2.1   Observational data

### 2.1.1   MERRA-2

The Modern-Era Retrospective analysis for Research and Applications, Version 2

(MERRA-2), represents the latest advancement in global atmospheric reanalysis during the
satellite era.  Produced by NASA's Global Modeling and Assimilation Office (GMAO), it utilizes
the Goddard Earth Observing System Model (GEOS) version 5.12.4 (Molod et al., 2015).  The
aerosol species are from the dataset, inst3_3d_aer_Nv, which is an instantaneous 3-dimensional 3-
hourly data collection in MERRA-2 (Modeling and Office, 2015).  The dataset comprises
assimilations of aerosol mixing ratio parameters at a native resolution of 0.5° latitude x 0.625°
longitude across 72 model layers, encompassing dust, sea salt, sulfur dioxide ($SO_2$), sulfate ($SO_4$),
black carbon (BC), and organic carbon (OC).  The data is provided every three hours, beginning
at 00:00 UTC.  Based on Wang et al. (2020), we also adopt MERRA-2 to drive the WRF-Chem
initial and boundary conditions for this study (see Sect. 2.2.2 for details).

### 2.1.2   Geostationary satellite retrievals (Meteosat)

Cloud properties are derived from the Spinning Enhanced Visible and Infrared Imager

(SEVIRI) on Meteosat-10 and Meteosat-11, which offer a spatial resolution of 3 km at nadir and
a half-hourly temporal resolution over the ENA region.  These SEVIRI cloud products are



generated using the Satellite ClOud and Radiation Property retrieval System (SatCORPS)
algorithms (Painemal et al., 2021). These methods, developed by the Clouds and Earth's Radiant
Energy System (CERES) project, are specifically tailored to support the ARM program at ARM
ground-based observation sites (Minnis et al., 2011; Minnis et al., 2021). Specifically, this study
adopts cloud fraction as the observational reference over the ENA region. The adopted data have
been specifically processed (e.g., solar zenith angle, cloud optical thickness, and cloud labels) and
averaged to 25 km × 25 km (Qiu et al., 2024).

### 161    2.1.3  Aircraft observation

The U.S. DOE ARM Aerosol and Cloud Experiments in the Eastern North Atlantic (ACE-

ENA) aircraft field campaign near the Azores islands provided extensive observations of the
vertical distributions of aerosol and cloud properties (Wang et al., 2022). Intensive operational
periods (IOPs) of the ACE-ENA took place in late June and July 2017, as well as January to
February 2018. During the 2017 summer IOP, the ARM Aerial Facility's (AAF) Gulfstream-159
(G-1) aircraft delivered precise measurements of aerosol size distribution, total aerosol number
concentration, and chemical constituents both below and above cloud layers. $SO_4$ and OC mass
concentrations were measured using the Aerodyne High-Resolution Time-of-Flight Aerosol Mass
Spectrometer (HR-ToF-AMS), while refractory BC was measured by the Single Particle Soot
Photometer (SP2). Detailed information about each instrument is available on the ARM website
(https://www.arm.gov/research/campaigns/aaf2017ace-ena). In this study, aircraft measurements
of $SO_4$, OC, and BC from 19 July 2017, are utilized to assess the simulated aerosol vertical profile.
However, uncertainties arising from the measurements and spatiotemporal sampling strategies
may hinder direct comparisons of absolute values between the observations and modeled results.



### 2.1.4 ARM ground-based observations


The DOE ARM ground-based instruments deployed on Graciosa Island in the Azores

archipelago provide comprehensive measurement of aerosols, clouds, radiation, atmospheric
boundary layer, and other atmospheric properties. In this study, liquid water path (LWP) is
retrieved from the brightness temperature measured by the microwave radiometer (MWR) at 23.8
and 31.4 GHz (Liljegren et al., 2001) and used for model evaluation. The temperature and moisture
profiles are from the interpolated sonde data, derived from the radiosonde measurement.

## 2.2 The model


### 2.2.1 WRF-Chem


The Weather Research and Forecasting (WRF) model version 4.4.2 (Skamarock et al.,

2021) coupled with a chemistry component (WRF-Chem) (Grell et al., 2005) is used in this study.
The standard WRF-Chem permits the simulation of the combined direct, indirect, and semi-direct
effects of aerosols (Grell et al., 2005; Fast et al., 2006; Chapman et al., 2009). WRF-Chem version
4.4.2 has sophisticated packages to represent chemistry processes (i.e., gas-phase reaction, gas-to-
particle conversion, coagulation, etc.) and aerosol size and composition (Binkowski and Shankar,
1995). In this study, the Regional Acid Deposition Model version 2 (RADM2) photochemical
mechanism (Stockwell et al., 1997) is integrated alongside the Modal Aerosol Dynamics Model
for Europe (MADE) and the Secondary Organic Aerosol Model (SORGAM) (Ackermann et al.,
1998; Schell et al., 2001) to simulate atmospheric chemistry and the evolution of anthropogenic
aerosols. MADE/SORGAM adopts a modal approach to represent the aerosol size distribution,
predicting mass and number concentrations across three aerosol modes (Aiken, accumulation, and
coarse). MADE/SORGAM has inorganic, organic, and secondary organic aerosols and contain
aerosol formation processes including nucleation, condensation, and coagulation. WRF-Chem

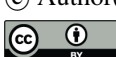



tracks the number of particles and the mass of chemical compounds (e.g., $SO_4^{2-}$, $NH_4^+$, $NO_3^-$, $Na^+$,
$Cl^-$ etc.) in each aerosol mode, including both interstitial aerosols and aerosols present in liquid
water (the sum of cloud and rain), as prognostic variables.

The size, composition, and mixing state of aerosols significantly influence their capability

to activate as CCN (Zaveri et al., 2010). A physically based aerosol activation parameterization
scheme has been developed for climate models to simulate CCN concentration accurately and
efficiently (Abdul-Razzak and Ghan, 2000). This aerosol activation parameterization was initially
designed for a single aerosol type with a lognormal size distribution. Then, they expanded this
parameterization to accommodate multiple externally mixed lognormal modes, with each mode
consisting of both soluble and insoluble materials internally mixed. However, WRF-Chem
(MADE/SORGAM) chemistry package adopts this global internal mixing assumption, where all
particles within a log-normal mode within the same grid cell are instantly combined, resulting in
the same chemical composition. This instantaneous internal mixing assumption modifies the
optical and chemical characteristics of particles in WRF-Chem simulations, potentially impacting
aerosol-cloud interactions, such as aerosol activation as CCN (Zhang et al., 2014).

### 214    2.2.2  The configuration

Our focus in this study is to examine aerosol-cloud interactions close to the scale of large-

eddy simulation (LES) over the ARM ENA site. We use WRF-Chem with a full chemistry
package involving sophisticated gaseous and aqueous chemical processing calculations and dry
and wet depositions. The numerical simulations are employed with 4 domains with 4 horizontal
resolutions of 5 km, 1.67 km, 0.56 km, and 0.19 km, respectively (Fig. 1), with one-way nesting.
Seventy-five vertically staggered layers are stretched to have a higher resolution near the surface
based on a terrain-following pressure coordinate system. With this setup, the model has roughly





24 model layers in the boundary layer (~2000 m). The time step is 30 and 10 seconds for advection
and physics calculation for the domains 1 and 2, respectively. The nesting inner domains 3 and 4
have the time step of 3 seconds and 1 second, respectively. The physics schemes adopted in the
simulations are listed in Table 1. The initial and boundary meteorological conditions are taken
from ERA5, developed by the Copernicus Climate Change Service (C3S) at ECMWF (European
Centre for Medium-Range Weather), stands as the fifth generation of ECMWF atmospheric
reanalysis, spanning from January 1940 to the present day (Hersbach et al., 2023). This
comprehensive dataset offers hourly estimates of numerous atmospheric, land, and oceanic climate
variables, covering the entirety of Earth on a 31km grid. The atmospheric component is resolved
using 137 levels, spanning from the surface up to 80 km in height.
The computational expense of conducting a 4-domain WRF-Chem simulation, particularly
with LES resolution, is exceedingly high. To mitigate this, we execute WRF solely for the two
outer domains (d01 and d02), leveraging the WRF downscaling module (ndown) (Skamarock et
al., 2008) to generate meteorological initial and boundary conditions for domain 3. As a result,
we only need to perform WRF-Chem simulations for the two inner domains (d03 and d04), leading
to an almost 50% reduction in total computational costs (compared to the original 4-domain run,
which had a throughput of 4 hours per day using 1080 cores). It is important to note that a high
temporal frequency for domain 3 boundary conditions is essential due to its fine horizontal
resolution (0.56 km). In this context, we update the boundary condition every 5 minutes for
domain 3.
To enhance the realism of aerosol mass simulation in remote marine regions, such as the
ENA site, we account for major aerosol species (BC, OC, and $SO_4$), as well as $SO_2$, from MERRA-
2 into the boundary conditions of the domain 3. Aerosols in the initial condition are introduced



into the restart file (wrfrst) following a one-hour initial run, rather than in the initial condition file
(wrfinput), to address certain numerical challenges. According to the emission setup for
MADE/SORGAM, we assume that the Aiken mode and the accumulation mode account for 20%
and 80% of the aerosol mass (BC and OC), respectively (Tuccella et al., 2012). Conversely, for
$SO_4$, 80% is allocated to the Aiken mode and 20% to the accumulation mode, reflecting the faster
growth rate of $SO_4$ and a longer duration of growth from the domain 3 boundary. Because
MERRA-2 only provides aerosol mass, the aerosol number concentrations for different aerosol
species are estimated with the density assumption of BC (1.7 g cm$^{-3}$), OC (1.0 g cm$^{-3}$) and $SO_4$
(1.77 g cm$^{-3}$) based on Liu et al. (2012).

It is common to consider that the ENA region is an unpolluted area because it is far away

from the anthropogenic pollution sources. Besides long-range transport aerosols, two local aerosol
sources, dimethyl sulfide (DMS) and sea salts, are also important for the aerosol budget. Kazil et
al. (2011) pointed that the observed DMS flux from the ocean in the VOCALS-REx field campaign
over the Southeast Pacific can support a nucleation source of aerosol. DMS oxidation by nitrate
($NO_3$) produces $SO_2$ and then increases $SO_4$ concentration (Toon et al., 1987). Since we adapted
$SO_2$ and $SO_4$ concentration from MERRA-2 in the initial and boundary conditions, we did not
double count DMS emissions in our simulations. As a result, chemical species emissions, except
for sea salt, are excluded from the simulations. The emission of sea salt particles is parameterized
using the method outlined by Clarke et al. (2006) in WRF-Chem. We adjust the parameter factor
for the sea salt emission to three times higher than the original estimate to achieve better agreement
with the sea salt aerosol variation in MERRA-2 reanalysis.



## 2.3 Study cases and numerical experiment design

We select three specific study cases to assess the impact of long-range transport aerosols on warm boundary layer clouds, with each case representing a typical meteorological regime observed over the ENA site. The first case, dated 1 July 2016, exhibits the formation of overcast stratocumulus clouds (Fig. 2a) within a meteorological regime characterized by a ridge system in the free troposphere and a high-pressure system near the surface (Fig. 2d). Predominant northwesterly and northerly winds in the area of the ARM ENA site coincide with the presence of long-range transport aerosols, commonly found along the periphery of the high surface pressure system (Logan et al., 2014; Gallo et al., 2023).

The second case on 19 July 2017 is a stratocumulus cloud case (Fig. 2b) within a post-trough regime featuring a high surface pressure under the influence of a trough system (Fig. 2e). Following the trough passage, robust northwesterly winds facilitated the influx of long-range transport aerosols into the region, which then shifted to northerly winds as the trough moved away. Because the ACE-ENA aircraft field campaign ran on this time, more aerosol observational data can be used to evaluate the model performance for this case.

Finally, the third case, dated 23 August 2019, occurred during a period of weak trough activity (Fig. 2f). Here, we noted the presence of broken, thicker clouds, often accompanied by deeper cloud formations (Fig. 2c). Long-range transport aerosols were again observed, primarily carried by northwesterly and northerly winds, albeit with weaker surface wind speeds compared to the preceding two cases.

All simulations start at 12 UTC on the preceding day of the study case, spanning a duration of 36 hours, with the initial 12 hours dedicated to spin-up. Again, aerosols in the initial condition

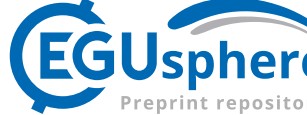



are introduced into the restart file after one-hour initial run (i.e., 13 UTC). The three
aforementioned cases, labeled as control cases (20160701_control, 20170719_control, and
20190823_control), are utilized to examine the behavior of warm boundary layer clouds under
diverse meteorological conditions.    Additionally, we formulated three perturbed cases
(20160701_perturbed, 20170719_perturbed, and 20190823_perturbed) by amplifying aerosol
concentrations in both initial and boundary conditions, as well as sea salt emissions, by a factor of
five relative to each control case. These control cases represent clean conditions, with near-surface
CCN concentrations below 100 cm$^{-3}$ at the ARM ENA site. A comparison between the control
and perturbed cases elucidates the sensitivity of warm boundary layer clouds to aerosol
enhancements under varying meteorological conditions, thereby contributing to a deeper
understanding of cloud microphysics processes under varying atmospheric dynamics.

## 299    3   Results

### 300    3.1   Model evaluation

#### 301    3.1.1   Meteorological conditions

Figures 2g, 2h, and 2i display the model-simulated liquid water path (LWP) in the control

runs over the domain 3 and 4. The simulations with fine spatial resolution effectively capture
synoptic frontal systems and cloud features, particularly when compared to the cloud images from
the Meteosat satellite (Figs. 2a, 2b, and 2c). Thin, uniform stratocumulus clouds on 1 July 2016
are simulated in 20160701_control, while the solid stratocumulus and frontal system on 19 July
2017 are also well captured in 20170719_control. Broken stratocumulus clouds on 23 August
2019 are reproduced in the simulation of 20190823_control.

The control runs serve as a basis for comparing the boundary layer structure against the

interpolated soundings obtained from the ARM ENA site. Figures 3 depict the comparison,



showing the simulated air temperature aligning closely with the observed values. However, on 1
July 2016, the model (20160701_control) displays a warm bias in capturing the temperature
inversion (Figs. 3a and 3b), with the simulated inversion layer situated approximately 200-300 m
lower than observed. Relative humidity has consistent performance and shows in Figs. S1a and
S1b. While the model indicates high relative humidity (> 90%) within 1000 m, observations show
this extending up to ~1200 m.

Moving to 19 July 2017, the model (20170719_control) successfully represents the diurnal

cycle of temperature vertical gradient within 1000 m height. However, compared to observations,
the model does not catch the inversion at 1500 m height and shows a warm bias in the model's
simulated temperature (Figs. 3c and 3d). The model simulation also tends to depict drier
conditions in the evening compared to the observation (Figs. S1c and S1d).

On 23 August 2019, characterized by a weak trough regime and higher boundary layer

height, the simulation of 20190823_control accurately captures warm and moist air advection in
the morning but struggles to maintain fidelity in the late afternoon. Notably, the lower troposphere
becomes excessively warm and dry after 18 UTC compared to observations (Figs. 3e, 3f, S1e, and
S1f).

In general, all simulations effectively capture large-scale conditions and cloud features

(Fig. 2) across different synoptic regimes but do not accurately represent temperature inversions
and air advection patterns. Discrepancies are noted in the simulated boundary layer height, which
is lower and the inversion is weaker than actual observations. Furthermore, the discrepancies tend
to increase in the later stage of simulation.



### 3.1.2 Aerosol evolution


As mentioned in Section 2.2.2, we incorporate major aerosol species (BC, OC, and $SO_4$),
from MERRA-2 into the domain 3 initial (in the restart file at 13 UTC) and boundary conditions
to enhance the realism of aerosol simulation. Figure 4 shows time-series $SO_4$ vertical profiles
from both MERRA-2 and WRF-Chem for three study cases. Here, we demonstrate the time
evolution of $SO_4$ because $SO_4$ is the main aerosol component among the three introduced aerosol
species, about 60~80% of total aerosol mass, in the initial conditions.
Compared to the MERRA-2 data, 20160701_control well captures the long-range transport
$SO_4$ between 1000 m and 2000 m, which is above the cloud deck, on 1 July 2016 (Figs. 4a and 4b).
The observed high BC and OC are also concentrated in this layer (Figs. S2a and S3a), as well as
simulated ones (Figs. S2b and S3b). Figure 4c and 4e show two MERRA-2 time-series vertical
distributions of $SO_4$ on 19 July 2017 and 23 August 2019, both showing low-altitude (below 1500
m) aerosol plumes. On 19 July 2017, the concentrations of BC and OC showed two peaks – one
near the surface and another above 1500 m in the free troposphere (Figs. S2c and S3c). This
pattern indicates the presence of a biomass-burning signature in the plume on that day (Wang et
al., 2020). While the simulation of 20170719_control did not capture the near surface BC, OC,
and $SO_4$ concentration after 12 UTC on 19 July 2017 (Figs. S2d, S3d, and 4d). It is because in the
case of the post-tough regime, the wind direction changes from northwesterly to northerly winds
when the trough moved away, the aerosol plume in the domain 3 did not propagate into the domain
4 when the wind direction change (figure not shown). However, the simulation of
20170719_control still captures the BC and OC plumes in the free troposphere (above 2000 m
height) (Figs. S2d and S3d).





Aircraft observations during the ACE-ENA provide more accurate depictions of aerosol
vertical distribution and aerosol layer heights, with differentiation of aerosol type.  Figure 5a shows
the vertical distribution of aerosol mass concentrations averaged over the flights on 19 July 2017.
BC, OC and SO$_4$ all increase with height above clouds (~1000 m), indicating downward
propagation of aerosol plumes and possible interaction with MBL clouds (600 – 1000 m).  Here,
we also see that high SO$_4$ in the free troposphere, same as the data in MERRA-2, but the model
underestimates the OC concentration in the free troposphere.  On the other hand, within the MBL,
there is a much higher concentration of SO$_4$ in the MBL than those of BC and OC in the
observations. This phenomenon is also captured by the WRF-Chem simulation (Fig. 5b), but the
model did not capture the magnitude of SO$_4$ concentration.
Similarly, for the case of 23 August 2019, within the low boundary layer, there is a much
higher concentration of SO$_4$ in the low boundary layer (Fig. 4e).  After 12 UTC on 23 August
2019, BC and OC show both high-altitude plumes and low-altitude plumes approaching into the
domain, which indicate potentially two different aerosol sources (Figs. S2e and S3e).  Again, while
the simulation of 20190823_control well captures the time evolution of aerosol plume, the
boundary of high-altitude plumes and low-altitude plumes appears 300 m lower in the simulations
(~600 m in altitude; Figs. S2f and S3f) compared to the observations (~900 m in altitude).
Sea salts serve as an important source of CCN over the ocean, particularly in unpolluted
conditions.  However, due to their larger particle size, sea salt particles tend to accumulate near
the ocean surface and are swiftly removed by dry deposition and sedimentation processes (Chin et
al., 2002).  As discussed in Section 2.2, we adjusted the parameter factor to three times its original
value to better align with the MERRA-2 dataset.  The simulation of 20160701_control (Figs. S4a
and S4b) accurately reproduces sea salt concentrations, both in magnitude and vertical distribution,





consistent with observations, same as the case of 20170719 (Figs. S4c and S4d). Nevertheless,
the model encounters difficulties in simulating sea salt concentrations for the case of 23 August
2019 (Figs. S4e and S4f), corresponding to a weak-trough system (Fig. 2c). Sea salt emissions in
WRF-Chem are driven by surface wind speed; however, the simulated surface wind speed matches
well with ERA-5 (Fig. S5). Hence, underestimated sea salt concentrations may stem from
inadequacies in emission parameterization, which is excessively determined by the surface wind
speed (Gong, 2003).
### 3.1.3 Cloud properties

In Fig. 6, we observe a comparison between the simulated results and observations of LWP

and CF at different spatial scales (4 km- and domain-average, respectively) to leverage the
spatiotemporal advantages offered by both sets of observations. The ARM ground-based
instrument recorded an LWP of over 400 g m$^{-2}$ during the nighttime with drizzles reaching to the
surface on 1 July 2016 (Fig. 6a). As the sunrise (around 6 UTC), the LWP decreases to a range
about 100 g m$^{-2}$, and then increases again to 600 g m$^{-2}$ after 22 UTC.

To compare with the ARM ground-based observations, the WRF-Chem simulated result is

averaged over 20 × 20 grids centered on the Azores, which corresponds to an approximate
resolution of 4 km (Fig. 6a). Overall, the model generates a thin cloud layer with an
underestimated LWP during the nighttime, capturing only 10-20% of the observed LWP. The
simulated clouds are more consistent with the observations during the daytime. However, it is
important to note that the LWP retrieved by MWR experiences significant uncertainties during
drizzling or precipitating conditions. This is primarily due to the scattering effects of large
raindrops and raindrops accumulating on the instrument's radome, which can result in an
overestimation of LWP (Tian et al., 2019; Cadeddu et al., 2020).





Figure 6b depicts the comparison of CF between observations and WRF-Chem. The CF
values obtained from Meteosat are close to 1, indicating a solid cloud field. In contrast, the CF
simulated by WRF-Chem range between 0.5 and 0.9 on a domain-averaged scale. Similar to the
LWP results, the simulated CFs from WRF-Chem exhibit a diurnal cycle, with higher values
during the nighttime and lower values during the daytime. Due to the thinner clouds simulated in
WRF-Chem based on LWP, the modeled CF is 40-60% lower than the observation in the
afternoon, indicating that clouds dissipate more quickly in the model.
Compared to a ridge system like the case of 20160701, the WRF-Chem model is harder to
capture warm boundary layer clouds under a regime characterized by a post-trough system (like
the case of 20170719) or a weak trough system (like the case of 20190823). Compared to the
observations, the simulated LWP in 20170719_control is about 30% of the observed value (Fig.
6c. In contrast, the simulated CF performs better, reaching about 75% of the observed value (Fig.
6d). The discrepancy between the modeled results and observations may arise from delayed
moisture transfer from the outer domain or insufficient vertical resolution. In this instance, the
cloud systems move quickly under the post-trough weather regime. A 5-minute moisture input
from the boundary condition using WRF downscaling (ndown) may not be sufficient to transport
moisture into the inner domain, making it difficult for the model to develop thicker marine
stratocumulus clouds, especially for such high spacing resolution. On the other hand, in another
ongoing project, we have observed that increasing the vertical layers to 99 levels significantly
enhances the simulated cloud amount (figure not shown). Another possible reason is that the 6th
Order Horizontal Diffusion used in the study (diff_6th_opt = 2) is easy to break down the marine
stratiform clouds, especially in the high spacing resolution (Knievel et al., 2007). It is worth to
mention that Christensen et al. (2024) conducted sensitivity tests using various shallow cumulus



and microphysics schemes, and the different combinations of these schemes had a substantial
impact on the simulated cloud amount as well.
Moving to the case of 20190823, overall, compared to the observations, the model captures
LWP and CF slightly better, especially in the domain-averaged scale (Figs. 6e and 6f). Based on
the LWP observed from ARM, there are two systems passing in the area, one between 7 UTC to
14 UTC and the other between 18 UTC to 24 UTC on 23 August 2019. The simulation of
20190823_control captures the first system, but a little bit underestimates LWP; however, the
model misses the second system. The model simulated CFs also match well with Meteosat (Fig.
6f). Only after 18 UTC, the model misses catching the second system. The CFs drops 50 – 70 %
compared to the observations.
The underestimation of the cloud layer from the model simulations results in insufficient
longwave cooling at the cloud top, which may contribute to a weaker boundary layer inversion
and a shallower boundary layer depth identified in the previous section (negative feedback) (Zheng
et al., 2021).

### 437   3.2  Aerosol composition and activation

The advantage of utilizing WRF-Chem to investigate aerosol-cloud interactions stems from
its capability to simulate the spatiotemporal distribution of CCN. This modeling is based on
various aerosol components and their sizes, as well as their dynamic responses to wet removal
processes associated with clouds and precipitation. In traditional simulations that rely on fixed or
prescribed aerosol distributions, accurately representing these factors can be particularly
challenging. WRF-Chem, however, allows for a more nuanced understanding by dynamically
modeling how aerosol populations evolve over time, especially after cloud evaporation processes.



During evaporation, the reduction in cloud water can lead to a re-entrainment of aerosols back into
the atmosphere, altering their concentration and properties. This change can affect subsequent
cloud formation and precipitation patterns, highlighting the importance of capturing these
interactions for reliable predictions.

In this section, we concentrate on aerosol activation, considering its size and chemical

composition across three different cases. The following section will discuss the aerosol indirect
effect and how changes in cloud properties feedback into the aerosol population and its activation
capability.

In Fig. 7a, the blue solid line and blue dashed line represent the vertical profiles of total

aerosol number concentration (including Aiken mode and accumulation mode) and aerosol number
concentration of the Aiken mode, respectively. These profiles are averaged over the domain 4 on
1 July 2016. The environment shown in the figure is characterized by its cleanliness, with a total
aerosol number concentration below the cloud top measuring less than 300 cm$^{-3}$. In the
20160701_control simulation, the total aerosol number is low, and approximately 70% of the total
aerosol numbers belong to the Aiken mode. According to the study conducted by Mccoy et al.
(2024), which utilized aerosol number concentration measurements from ARM airborne
observations on 15 July 2017, it was found that the ratio of the Aiken mode to the total aerosol
number was approximately 50-60% within an altitude of 1000 m. Compared to this observational
analysis, our simulations generate an overabundance of small-sized aerosols, which result in a low
concentration of CCN. This discrepancy arises from the assumptions made when constructing the
aerosol initial and boundary conditions, which is the assumptions regarding the aerosol mode ratio
of SO$_4$ (80% for Aiken mode and 20% for accumulation mode).





The CCN calculation presented in Fig. 7a is based on the Köhler theory, which considers
both the aerosol size (curvature effect) and the chemical composition (solution effect) to estimate
the theoretical CCN number concentration at different supersaturations.    Under 1.0%
supersaturation, the CCN number concentration is found to be 42% of the total number of aerosol
number (could be estimated from 100% of accumulation mode and 16% of Aiken mode) (Fig.
S6a).  In the simulation of 20160701_control, the CCN number concentration under 0.2% (0.5%)
supersaturation is only 11% (25%) of the total aerosol number, which is lower than the
observations reported in Wang et al. (2020), where the observed CCN number concentration under
0.35% supersaturation was approximately 25% of the total aerosol number.  Even though $SO_4$ is
the dominant chemical component, accounting for nearly 50% (as shown in Figs. 7b and 7c), the
presence of an excessive number of Aiken mode aerosols may be the primary reason for the low
activation rate.  The curvature effect caused by these Aiken mode aerosols hinders their ability to
act as CCN effectively.
In the simulation of 20170719_control, the most aerosols within a height of 1000 m, which
is also the cloud layer (Fig. 7d).  The average aerosol number concentration across the entire
domain is measured to be 1286 $cm^{-3}$ within a height of 2000 m and the Aiken mode is 80% of the
total aerosol number in this case.  The chemical composition of aerosols in the 20170719_control
mainly is $SO_4$, and "others" (like sea salts) is second (Figs. 7e and 7f).  This variation in vertical
distribution leads to more aerosols being activated under the cloud top at a height of 1500 m.  This
is attributed to the presence of a peak value of accumulation mode aerosols and $SO_4$ at this height.
Because of high Aiken mode aerosols in this simulation, overall, the activation rate is low.
Under 1.0% supersaturation, the CCN number concentration is estimated to be 25% of the total
aerosol number.  This could be a result of 100% of the accumulation mode aerosols and 7% of the

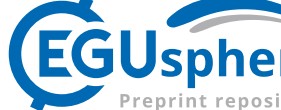

Aiken mode aerosols contributing to the CCN population (Fig. S6c). The CCN number
concentration under 0.2% (0.5%) supersaturation is only 4% (12%) of the total aerosol number.

Among the three cases studied, the case of 20190823 stands out as the most polluted case,

but the aerosol component and vertical distribution are close to the case of 20170719 (Figs. 7c and
7i). The average aerosol number concentration across the entire domain is measured to be 1850
$cm^{-3}$ within a height of 2000 m. The Aiken mode aerosols are also high and contribute to more
than 75% of the total aerosol number in this case (Figs. 7g and S6e). High $SO_4$ component also
leads to more aerosols being activated under the cloud top at a height of 2000 m. The CCN number
concentration under 0.2% (0.5%) supersaturation is only 6% (17%) of the total aerosol number,
slightly better than the case of 20170719.

The cloud droplet numbers observed in the three cases fall within the range of CCN

numbers under 0.1% and 0.2% supersaturation. Therefore, in the subsequent sections, we utilize
the CCN number concentration under 0.2% supersaturation as a representative of the CCN
activation rate.
## 3.3  Cloud responses to aerosol perturbations

Figure 8 illustrates the comparison of time series profiles of cloud water content (CWC)

and CCN number concentration under 0.2% supersaturation between the control runs and
perturbed runs. This figure also demonstrates the CCN spatiotemporal variation in our
simulations. Specifically, for the case of 20160701, it is evident that the CWC in
20160701_perturbed exhibits a positive response to increased CCN compared to the CWC in
20160701_control. This result aligns with the most of WRF studies that use fixed or prescribed
CCN numbers to investigate aerosol-cloud interactions (Wang et al., 2020; Christensen et al.,

2024).

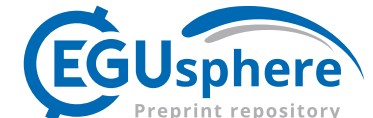

Figure 9a depicts the time series of domain-averaged LWP, encompassing both cloud and

rain, and CCN number concentration under 0.2% supersaturation for both the 20160701_control

and 20160701_perturbed cases. This visualization provides a quantitative representation of the

change in CCN number concentration, which increases from a mean value of 32.52 cm$^{-3}$ in the

control run to 127.68 cm$^{-3}$ in the perturbed run, approximately three times higher than the control

run. Because we want to avoid counting high CCN number concentration above cloud top which

are also hard to become cloud droplets, the CCN number concentration is averaged within 1000 m

height (Wang et al., 2020).

The LWP in the 20160701_control case exhibits a domain mean value of 64.88 g m$^{-2}$,

which subsequently increases to 123.27 g m$^{-2}$ in the 20160701_perturbed case. As mentioned in

Section 3.1, the LWP for the 20160701 case follows a diurnal cycle, with higher values during

nighttime and lower values during daytime. This diurnal cycle is also observed in the perturbed

simulation, with the larger differences in CCN and LWP between the control run and perturbed

run during nighttime (Fig. 9a).

After increasing aerosols, the cloud droplet number in the 20160701_perturbed run

demonstrates similar responses. In the 20160701_control case, the domain mean value of cloud

droplet number is 14.03 cm$^{-3}$, which subsequently increases to 45.52 cm$^{-3}$ in the

20160701_perturbed case. As the cloud droplet number increases, the cloud radius decreases from

12.23 μm in the control run to 10.08 μm in the perturbed case.

The case of 20170719 represents a post-trough weather regime, and Fig. 8c illustrates the

passage of a frontal system in the area after 9 UTC on that day. In the 20170719_perturbed

simulation, the CWC increases following the system's passage (Fig. 8d) compared to the CWC in

the 20170719_control run. Additionally, the ambient CCN number in the perturbed run is also





higher. The time variation of CCN concentration in Fig. 9c shows elevated CCN numbers before
and after the system enters the domain. In the 20170719_control case, the domain mean value of
CCN number concentration is 60.51 cm$^{-3}$, which subsequently increases to 253.51 cm$^{-3}$ in the
20170719_perturbed case. The domain-averaged LWP also exhibits an increase, rising from 59.31
g m$^{-2}$ in the 20170719_control run to 74.07 g m$^{-2}$ in the 20170719_perturbed case. Notably, this
change primarily occurs after the passage of the frontal system.

The cloud droplet number consistently shows higher values in the perturbed case (Fig. 9d),

and this pattern is similar to the difference in CCN between the two runs of 20170719 (Fig. 9c).
In the 20170719_control case, the domain mean value of cloud droplet number is 20.70 cm$^{-3}$, while
the value is 56.09 cm$^{-3}$ in the perturbed case. When the cloud droplet number increases in the
perturbed run, the cloud radius decreases from 9.90 μm in the control run to 7.49 μm in the
perturbed case. This reduction in cloud radius is even smaller than the cloud radius observed in
the case of 20160701.

The case of 20190823 is similar to the case of 20170719, but it represents a weak trough

weather regime. Figure 8e also illustrates the passage of a cloud system in the area between 6
UTC to 18 UTC on 23 August 2019, and the CWC in the perturbed run increases during this period.
Quantitatively, in the 20190823_control case, the domain mean value of CCN number
concentration is 124.32 cm$^{-3}$, which subsequently increases to 475.37 cm$^{-3}$ in the
20190823_perturbed case, which is also about three times higher. The domain-averaged LWP
also exhibits an increase, rising from 48.92 g m$^{-2}$ in the 20190823_control run to 58.53 g m$^{-2}$ in
the 20190823_perturbed case.

Differ from the case of 20170719, the frontal system moved away from the study domain

after 12 UTC, the differences of CCN number or cloud droplet number between the control and



perturbed runs becomes even more pronounced after the system (Figs. 9e and 9f).  In the
20190823_control case, the domain mean value of cloud droplet number is 33.94 cm$^{-3}$, while the
value is 79.97 cm$^{-3}$ in the perturbed case.  When the cloud droplet number increases in the
perturbed run, the cloud radius decreases from 8.51 μm in the control run to 6.45 μm in the
perturbed case.  This reduction in cloud radius is similar to the cloud radius observed in the case
of 20170719.

We observe that large aerosol-induced LWP occurs during the periods of rainfall (Fig. S7).

To accurately quantify the differences, we calculate the average LWP over approximately 25 km
of the domain 4.  This results in 16 averaged grids per output file, with each file generated every
10 minutes.  This averaging process is based on Arola et al. (2022) and Zhou and Feingold (2023)
to avoid the impact of heterogeneity and co-variability on the results.  Specifically, we aggregate
the simulation grids with a spacing resolution of approximately 190 m to form a larger grid of
around 25 km for each 10-minute simulation output, as presented in Table 2.

Table 2 presents the 10-minute mean and standard deviation of several variables, including

CCN, LWP, cloud droplet number (Nc), cloud radius (Re), and rainfall intensity (RI), across three
study cases.  The classification of "rain" and "non-rain" is based on the RI (unit: mm hr$^{-1}$) on the
averaged grid.  Specifically, a grid is considered as "rain" if the RI is greater than zero.  In the
control cases, the averaged CCN number is 73.07 cm$^{-3}$, and the corresponding LWP is 53.17 g m$^{-2}$
.  However, in the perturbed cases, the CCN number increases approximately threefold, reaching
218.21 cm$^{-3}$, and the LWP increases by 49% to 79.25 g m$^{-2}$.  The introduction of additional aerosols
in the perturbed cases also leads to a significant increase in the Nc number, from 22.68 cm$^{-3}$ in the
control cases to 59.74 cm$^{-3}$ in the perturbed cases.  Consequently, the Re decreases by 21% from
9.97 μm to 7.83 μm, and the RI decreases by 11% from 0.009 mm hr$^{-1}$ to 0.008 mm hr$^{-1}$.



To investigate the interaction between aerosols and clouds, we analyze the results
separately for rain and non-rain grids. In both the control and perturbed cases, we observe that the
CCN number within 1000 m is lower in the rain grids compared to the non-rain grids, primarily
due to the washout effect caused by rainfall. Additionally, the LWP over the rain grids is generally
higher than that over the non-rain grids. Furthermore, when comparing the control and perturbed
cases, we find that the LWP over the rain grids increases by 57% from 58.57 g m$^{-2}$ to 91.81 g m$^{-2}$.
In contrast, the LWP over the non-rain grids only increases by 28% (Table 2). This difference can
be attributed to the conversion of cloud droplets to raindrops through processes like autoconversion
and collection, which occurs more prominently over the rain grids. We also observe that in the
non-rain grids, especially at the cloud edges (or low LWP), the perturbed cases reveal an increased
presence of small cloud droplets. This abundance of smaller droplets facilitates evaporation,
resulting in a reduced LWP (e.g., clouds in the bottom right corner of Figs. S8a and S8b).
Consequently, the Nc number over the rain grids is lower compared to the Nc number over the
non-rain grids. Moreover, when introducing aerosols in the perturbed runs, the results over the
rain grids exhibit larger cloud drops and a wider radius spectrum compared to the results over the
non-rain grids. This suggests that the presence of aerosols has a more pronounced effect on cloud
properties within the rain grids.
Zheng et al. (2022) conducted a study on the aerosol-cloud interaction using ground-based
measurements from the ARM program, focusing on the influence of environmental variables.
Their findings revealed that when there is ample water vapor and low CCN loading, the active
coalescence process leads to a broader size distribution of cloud droplets, resulting in an increase
in cloud droplet radius. On the other hand, when there is enhanced activation of CCN and
condensational growth of cloud droplets due to higher CCN loading below the cloud, the cloud





droplet radius decreases. This combined effect signifies an intensified aerosol-cloud interaction,
leading to a broad range of cloud droplet radii. The simulated results in our study, specifically
over the rain grids where a sufficient water vapor environment is considered, demonstrate a
significant aerosol-cloud interaction, where increased CCN introduces more newly converted
droplets, resulting in a broad range of cloud droplet radii.
Since we utilize a comprehensive aerosol module in WRF-Chem to examine aerosol-cloud
interactions, we are able to explore how changes in cloud properties, driven by increased CCN,
affect aerosol concentrations. For example, in the post-trough regime (20170917 case) and the
weak trough regime (20190823 case), we observe that the cloud structure exhibits more open-cell
stratocumulus clouds (Figs. 2h and 2i). As motioned above, the increased number of smaller cloud
droplets at the cloud edge facilitate evaporation and results in a lower LWP (Fig. S8). The larger
aerosols from the evaporated clouds return to the accumulation mode, making them more likely to
activate as CCN again.
To demonstrate how robust this process on the ACI, we calculate the time series of the
ratio of the percentage of activated CCN at 0.2% supersaturation to the accumulation mode
aerosols between perturbed and control runs, defined
as $\left.(CCN_{0.2\%}/Accu.\,aerosols)_{perturbed}\middle/(CCN_{0.2\%}/Accu.\,aerosols)_{control}\right.$, shown in Fig. 10. A
ratio greater than 1 suggests that accumulation mode aerosols in the perturbed cases are more
readily activated as CCN at a supersaturation of 0.2%, especially in the cases of 20170719 and
20190823. Conversely, a ratio less than 1 is observed during the first half of the day in the
20170701 case, which is attributed to the very low levels of accumulation mode aerosols in the
model.





### 3.4 Cloud liquid water path (LWP) susceptibilities


In this study, the susceptibility of LWP to changes in CCN concentration is quantified

using the logarithmic slope between LWP and CCN, denoted as $dln(LWP)/dln(CCN)$
(Gryspeerdt et al., 2019). This slope represents the sensitivity of warm stratocumulus clouds' LWP
to variations in CCN concentration, like shown in Fig. S8c. As presented in Table 2, we aggregate
the simulation grids with a spacing resolution of approximately 190 m to form a larger grid of
around 25 km for each 10-minute simulation output. This averaging process helps to reduce the
impact of heterogeneity and co-variability on the results.

Figure 11 illustrates the averaged cloud susceptibilities for various LWP and CCN or Nc

bins across three study periods. The logarithmic slope between LWP and CCN is calculated at
each output time (every 10 minutes) using data from 16 aggregated grid points from the control
run and 16 aggregated grid points from the perturbed run. Our study reveals that when the CCN
concentration is below 100 cm$^{-3}$, the susceptibility for different LWP and CCN values is positive
and the values are large, indicating that changes in LWP are sensitive to variations in CCN number.
Here also demonstrates the AIE is large when an increase in CCN can have a large impact on LWP
enhancement. However, when the mean CCN concentration exceeds 100 cm$^{-3}$, the LWP
susceptibility becomes small, both positive and negative. This suggests that the change in LWP is
not as sensitive to changes in CCN number (as shown in Fig. 11a). It is important to note that the
CCN number used in our study is averaged within a 1000 m range, which may introduce
uncertainty to the absolute values of susceptibility by including aerosols that are not directly
involved in the aerosol-cloud interaction (Wang et al., 2020).

Additionally, our simulations indicate that the Nc in this study is generally low, with a

mean value typically below 80 cm$^{-3}$. For different LWP and Nc values, the susceptibility is mostly



positive, indicating that changes in LWP are sensitive to variations in Nc number (as shown in Fig.
11c).

When we investigate the variation of LWP susceptibility over time, we observe that

positive susceptibilities for different LWP and CCN (Nc) typically occur during periods of no rain
or light rain (Figs. 12 and S7). On 1 July 2016, the time series of LWP susceptibility for different
CCN or Nc shows a diurnal cycle, with large positive values during the nighttime and small
positive values in the afternoon. During heavy rain events, such as from 9 to 16 UTC on 19 July
2017 and from 0 to 15 UTC on 23 August 2019 (Fig. S7), the LWP susceptibilities are negative or
close to zero (Figs. 12c and 12e). In the perturbed cases, during the heavy rainfall periods, some
aggregated grids show very low LWP (Fig. S8c). This reduction in LWP is caused by the
evaporation from small cloud droplets at non-rain grids on the cloud edge. Those low LWP grids
in the perturbed runs result in a negative or near-zero logarithmic slope between LWP and CCN
(Figs. S8c, 12c and 12e), although the domain mean LWP is higher in the perturbed case than in
the control case (Figs. 9c and 9e).

To further illustrate the reduction in LWP due to evaporation at the cloud edge, Fig. 13

presents the relative change in $ln(LWP)$ between the perturbed and control cases across different
LWP percentile ranges in the control case during the periods of negative LWP susceptibility, as
shown in Fig. 12. The results indicate a decrease in LWP in the perturbed cases compared to the
control cases for pixels with the lowest LWP percentile range (0-25%), which we assume occur at
the cloud edges.

Figures 11b and 11d display the mean Re susceptibilities for different CCN and Nc,

respectively. The results consistently show that as CCN or Nc increases, the radius of the cloud



droplets decreases. Additionally, the change in Re is more pronounced when the Nc (or CCN) is
higher.

# 674    4   Discussion and summary

This study focuses on aerosol indirect effects (AIE), particularly involving long-range

transport aerosols, in the Eastern North Atlantic (ENA) region. It specifically examines these
effects on warm boundary layer stratiform clouds located on the eastern side of oceanic subtropical
highs under three different weather regimes: a ridge with a surface high-pressure system, a post-
trough with a surface high-pressure system, and a weak trough. We select three specific study
cases (i.e., 20160701, 20170719, and 20190823) to assess the impact of long-range transport
aerosols on warm boundary layer clouds, with each case representing a typical meteorological
regime observed over the ENA site.

To investigate aerosol-cloud interactions more realistically, incorporating aerosol

chemistry components that activate to cloud condensation nuclei (CCN) and accounting for aerosol
spatiotemporal variation, this study employs the Weather Research and Forecast model coupled
with a chemistry component (WRF-Chem). This approach provides a detailed examination of AIE
in the ENA region under the three specified weather regimes. We employ a downscaling technique
to conduct WRF-Chem simulations for the two inner domains (with the outer domains utilizing
WRF). This approach results in nearly a 50% reduction in total computational costs, achieving a
throughput of 8 hours per day using 1,080 cores.

We incorporate major aerosol species (BC, OC, and $SO_4$), as well as $SO_2$, from MERRA-

2 to provide aerosol initial and boundary conditions, labeled as control cases. Additionally, we
formulate three perturbed cases by amplifying aerosol concentrations in both initial and boundary
conditions, as well as sea salt emissions, by a factor of five relative to each control case. Since





aerosol features are primarily determined by aerosol initial and boundary conditions, a higher
Aiken mode assumption in the major aerosol component (i.e., $SO_4$) regarding the aerosol mode
ratio (80% for Aiken mode and 20% for accumulation mode) results in fewer aerosols activating
as CCN due to the curvature effect in our simulations.

The WRF-Chem model captures the cloud structure in the case of 20160701. It simulates

the formation of thin, uniform stratocumulus clouds within a meteorological regime characterized
by a ridge system in the free troposphere and a high-pressure system near the surface. However,
the cases of 20170719 and 20190823 exhibit the development of broke and thicker solid
stratocumulus clouds within a post-trough regime and a weak trough, respectively. With the fast-
moving cloud systems and strong surface wind, the WRF-Chem model struggle to capture the
development and movement of these cloud systems due to delayed moisture transport from outer
boundary condition and potential insufficient vertical resolution.

In all cases, compared to the observations, the WRF-Chem model underestimates the liquid

water path (LWP) and cloud fraction due to warmer and lower simulated boundary layer. In the
perturbed cases, we find 57% higher aerosol-induced LWP, especially during the periods of
rainfall. We also note that the perturbed cases exhibit lower rainfall intensity, indicating a rainfall
suppression effect attributed to high CCN concentrations as concluded in previous studies (Wang
et al., 2020; Christensen et al., 2024). In contrast, the LWP over the non-rain grids only increases
by 28%. Moreover, when introducing aerosols in the perturbed runs, the results over the rain grids
exhibit larger cloud drops and a wider radius spectrum compared to the results over the non-rain
grids. This suggests that the presence of aerosols has a more pronounced effect on cloud properties
within the rain grids. The non-rain grids over the cloud edge can have lower LWP because smaller
cloud droplets are easy to evaporate.



Our study further elucidates the intricate feedback mechanisms governing aerosol-cloud
interactions and aerosol properties. In both the post-trough and weak trough regimes, we observe
a pronounced tendency for the cloud structure to develop more open-cell stratocumulus clouds.
At the peripheries of these clouds, the perturbed cases demonstrate a significant increase in the
presence of small cloud droplets. This heightened abundance of smaller droplets not only
promotes evaporation but also leads to a marked reduction in LWP.
As these clouds evaporate, the larger aerosols that are released return to the accumulation
mode. This transition enhances their likelihood of reactivating as CCN. Consequently, this cycle
underscores the dynamic interplay between aerosol properties and cloud formation, highlighting
how changes in aerosol concentrations can influence cloud microphysics and, ultimately,
precipitation processes.
Additionally, the susceptibility of LWP to changes in CCN concentration is quantified
using the logarithmic slope between LWP and CCN. Our result shows when the CCN
concentration is low, LWP is sensitive to variations in CCN number, with higher CCN number
concentration leading to higher LWP. However, when the mean CCN concentration is relatively
high, LWP is not as sensitive to changes in CCN, the LWP susceptibilities are small in magnitude,
with both positive and negative values. Those negative values are caused by the evaporation from
small cloud droplets at non-rain grids on the cloud edge.
In Wang et al. (2020), the LWC susceptibility for a light precipitation case on 18 July 2017
also shows positive values based on three sensitivity runs with CCN concentrations of 10, 100,
and 1000 cm$^{-3}$. The cloud properties in their study are averaged over all cloud points in the
innermost domain. We adopt the same method as Wang et al. (2020) to estimate the LWP
susceptibility        using        the        domain        mean        values,        defined        as



$\Delta ln(LWP_{perturbed} - LWP_{control})/\Delta ln(CCN_{perturbed} - CCN_{control})$. This approach
predominantly yields positive values for LWP susceptibility across the three study cases (see Figs.
S9 and S10). This suggests that the 25-km resolution is critical for accurately estimating LWP
susceptibility. A resolution that is too coarse (i.e., using the domain mean) may fail to capture
finer details, such as the aerosol drying effects occurring at the cloud edges.
Conversely, the LWP susceptibilities associated with varying cloud droplet numbers
reported in Qiu et al. (2024) reveal significant negative values in LWP susceptibility in response
to high cloud droplet numbers, a trend that is partially reflected in our study. Further investigation
is required to reconcile the difference in LWP responses between observational data and model
simulations. Additionally, a more accurate estimation of LWP susceptibility to changes in CCN
concentration is necessary.

**Code and data availability:**
The WRF-Chem code (v4.4.2) used in this study has been released on GitHub
(https://github.com/wrf-model/WRF/releases/download/v4.4.2/v4.4.2.tar.gz). The observational
data used in this study are available at https://doi.org/10.5281/zenodo.13356995. Other WRF-
Chem simulated outputs for the plots in this paper are available at
https://doi.org/10.5281/zenodo.13357040.

**Author contributions:**



H.-H. Lee and X. Zheng provided ideas and designed the experiments in this study. H.-H. Lee
conducted all the simulations and analyses. H.-H. Lee leads and coordinates the manuscript with
inputs from coauthors.

**Competing interests.**
At least one of the (co-)authors is a member of the editorial board of Atmospheric Chemistry and
Physics.

**Acknowledgements:**
This work is supported by the DOE Office of Science Early Career Research Program and the
ASR Program. Work at LLNL was performed under the auspices of the U.S. DOE by Lawrence
Livermore National Laboratory under contract DE-AC52-07NA27344. LLNL IM: LLNL-JRNL-
870513-DRAFT.

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




Table 1. WRF physics scheme configuration

| Physics Processes | Scheme | Reference |
|---|---|---|
| Microphysics | Morrison (2 moments) scheme | Morrison et al. (2009) |
| Longwave radiation | RRTMG scheme | Mlawer et al. (1997) |
| Shortwave radiation | RRTMG scheme | Iacono et al. (2008) |
| Surface-layer | Monin-Obukhov surface layer | Monin and Obukhov (1954) |
| Land surface | Unified Noah land-surface model | Chen and Dudhia (2001) |
| Planetary boundary layer | MYJ (Eta) TKE scheme (d01 and d02 only) | Mellor and Yamada (1982) Janjić (1994) |
| Shallow cumulus parameterization | GRIMS scheme (d01 and d02 only) | Hong and Jang (2018) |






Table 2. Ten-minute mean and standard deviation of cloud condensation nuclei (CCN), liquid
water path (LWP), cloud droplet number (Nc), cloud radius (Re), and rainfall intensity (RI) over
three study cases. Data are averaged over ~25 km of the domain 4 and total 16 averaged grids are
in the domain 4. Rain and non-rain are averaged the grids when the RI on the grid is larger than
and equal to zero, respectively. Only CCN are averaged within 1000 m height over the domain 4,
other variables are averaged within 2000 m height.

| Area | Case | CCN (cm⁻³) | LWP (g m⁻²) | Nc (cm⁻³) | Re (µm) | RI (mm hr⁻¹) |
|---|---|---|---|---|---|---|
| Domain | Control | $73.07 \pm 48.77$ | $53.17 \pm 32.65$ | $22.68 \pm 11.59$ | $9.97 \pm 2.31$ | $0.009 \pm 0.033$ |
| | Perturbed | $286.88 \pm 183.69$ (+293%) | $79.25 \pm 56.62$ (+49%) | $59.74 \pm 27.29$ (+163%) | $7.83 \pm 2.02$ (-21%) | $0.008 \pm 0.033$ (-11%) |
| Rain | Control | $68.15 \pm 48.05$ | $58.57 \pm 31.69$ | $20.17 \pm 9.33$ | $10.47 \pm 2.07$ | $0.011 \pm 0.035$ |
| | Perturbed | $250.14 \pm 153.23$ (+267%) | $91.81 \pm 55.06$ (+57%) | $53.01 \pm 20.39$ (+163%) | $8.35 \pm 1.83$ (-20%) | $0.009 \pm 0.036$ (-18%) |
| Non-Rain | Control | $103.73 \pm 41.52$ | $18.91 \pm 9.81$ | $38.57 \pm 11.93$ | $6.81 \pm 0.76$ | $0 \pm 0$ |
| | Perturbed | $444.47 \pm 217.08$ (+328%) | $24.22 \pm 15.80$ (+28%) | $89.24 \pm 33.42$ (+131%) | $5.54 \pm 0.90$ (-19%) | $0 \pm 0$ |




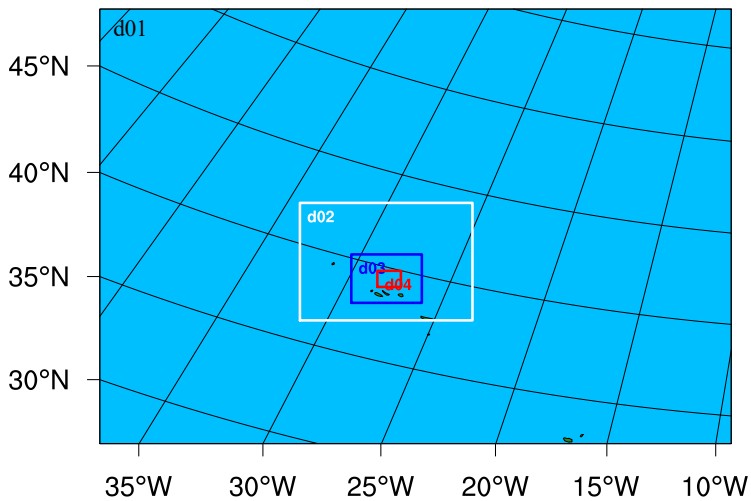


Figure 1. Model domains are designed for simulations. The 4 domains with 4 horizontal
resolution of 5 km (d01), 1.67 km (d02), 0.56 km (d03), and 0.19 km (d04), respectively.



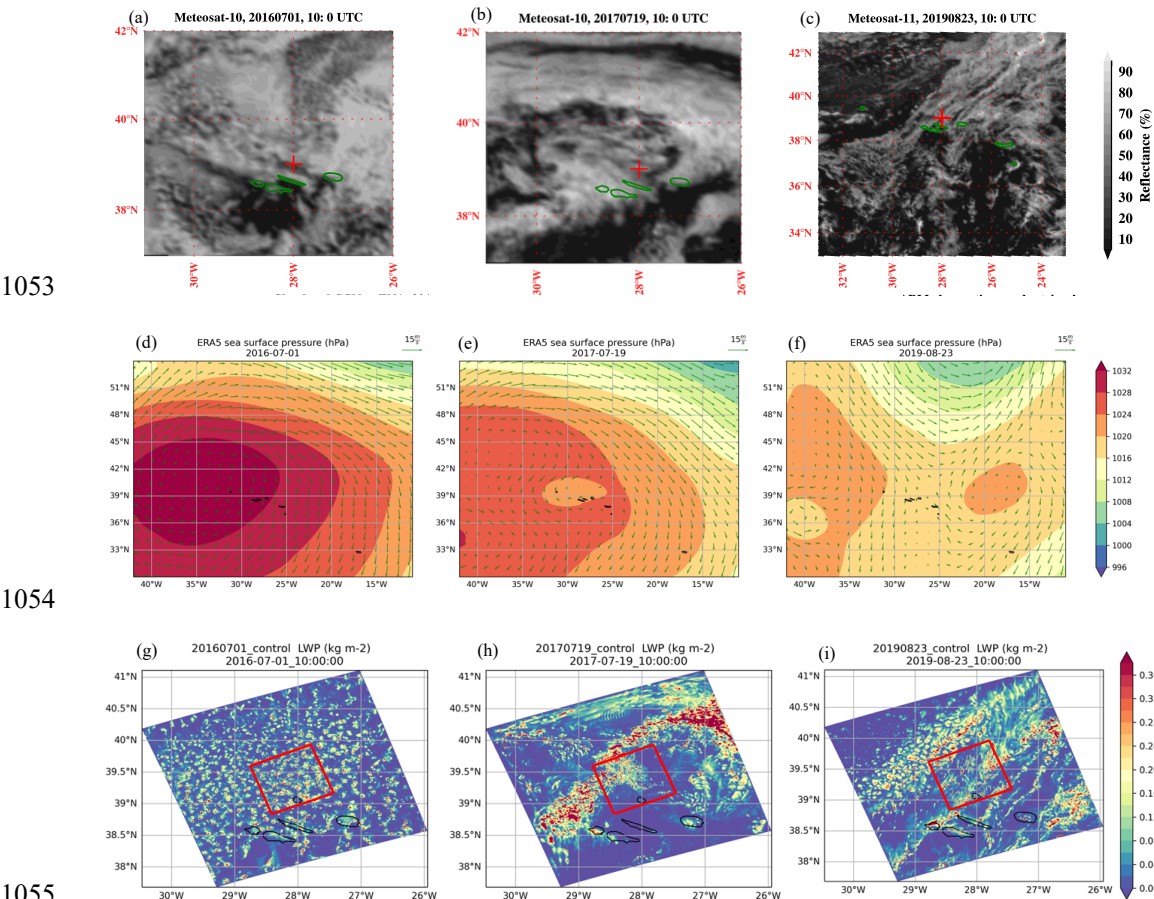




Figure 2. Spinning Enhanced Visible Infra-Red Imager (SEVIRI) images from Meteosat satellite at 10:00 UTC on (a) 1 July 2016, (b) 19 July 2017, and (c) 23 August 2019 over the ENA. (d), (e), and (f) are on the same day of (a), (b), and (c), respectively, but they are from ERA5 mean sea surface pressure (contour; units: hPa) and 10-meter surface wind (arrow; units: m s$^{-1}$). (g), (h), and (i) are on the same day of (a), (b), and (c), respectively, but they are WRF-Chem simulated liquid water path (LWP; units: kg m$^{-2}$) in the control runs. The red boxes in the figures indicate the result from the domain 4.

1063

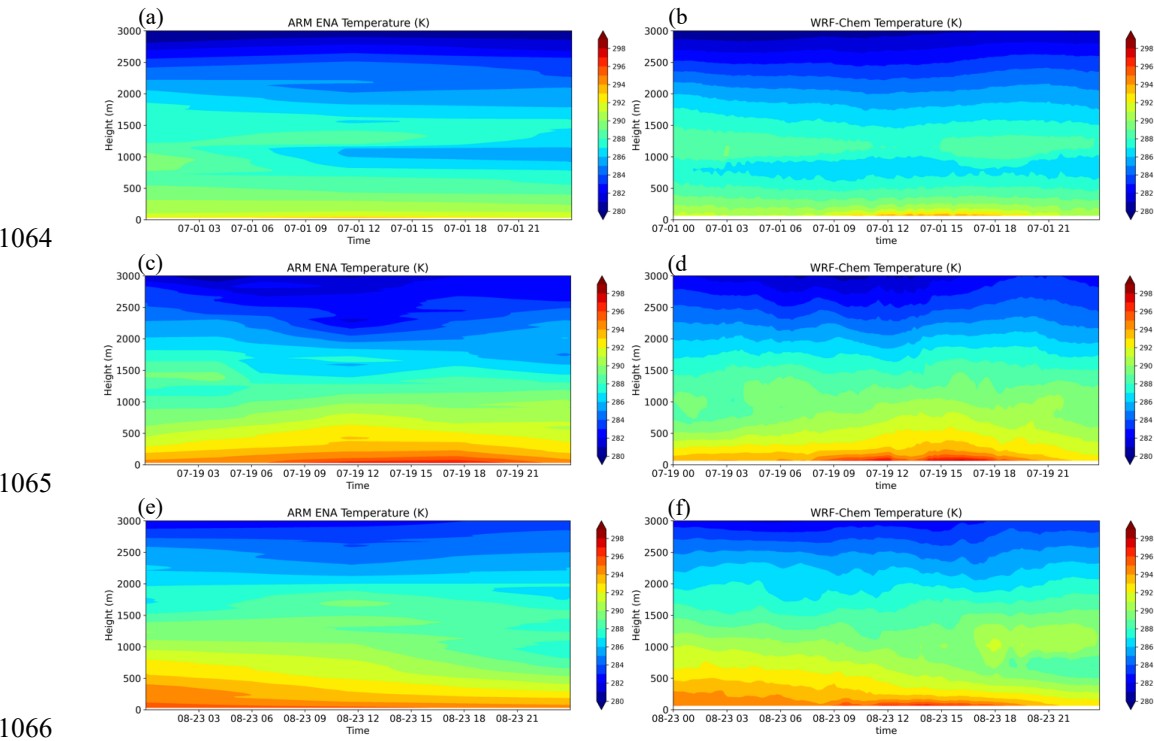

Figure 3. The time series of temperature profiles (units: K) from ARM interpolated soundings at the Azores (39.09°N, -28.02°W) on (a) 1 July 2016, (c) 19 July 2017, and (e) 23 August 2019. Panels (b), (d), and (f) depict the same dates as (a), (c), and (e), respectively, but show the average temperature from WRF-Chem simulated results over 20 × 20 grids centered on the Azores (approximately 4 km resolution).

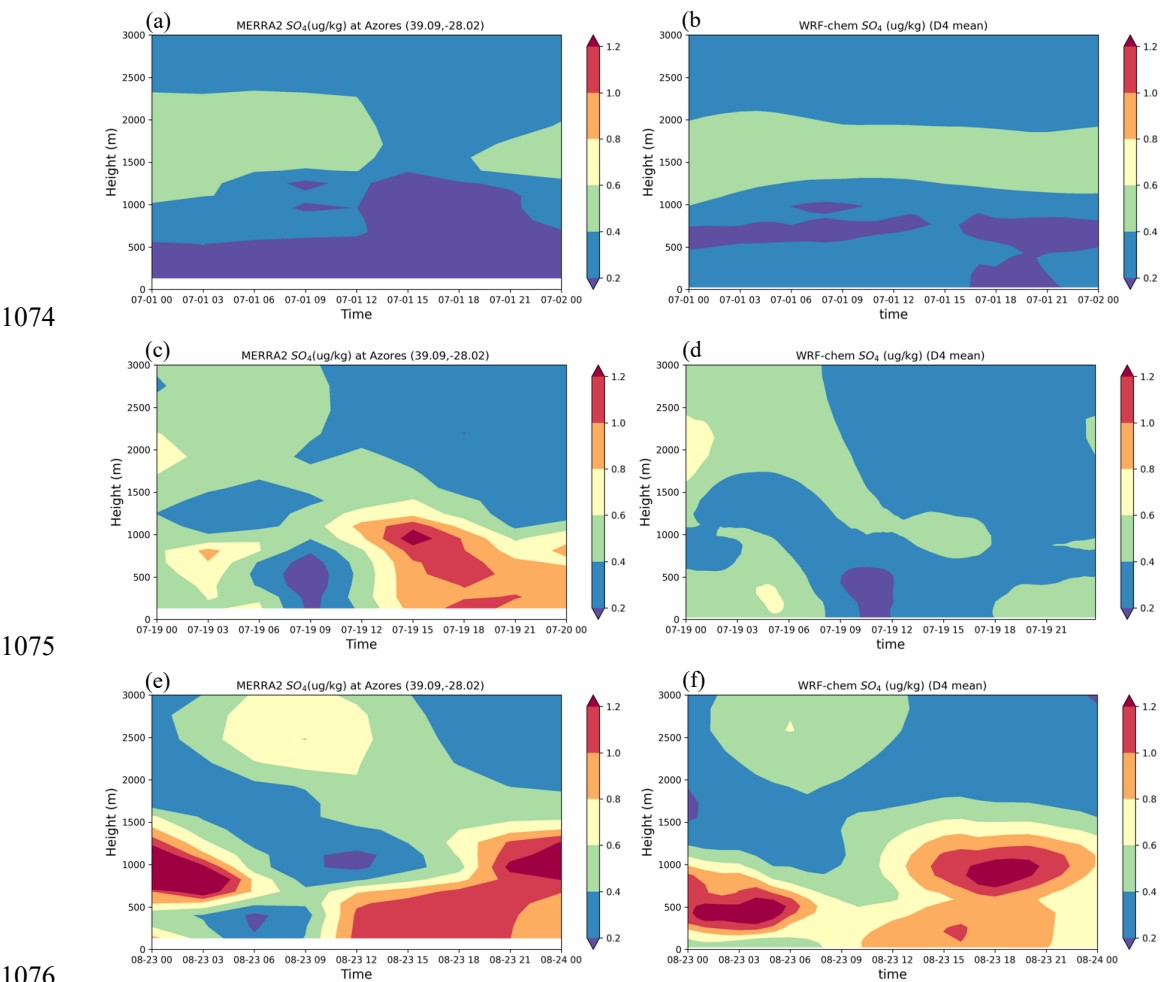

1074

1075

1076

Figure 4. The time series of SO$_4$ profiles (units: µg kg$^{-1}$) from MERRA-2 at the Azores
(39.09°N, -28.02°W) on (a) 1 July 2016, (c) 19 July 2017, and (e) 23 August 2019. Panels (b),
(d), and (f) depict the same dates as (a), (c), and (e), respectively, but show the average aerosol
concentration from WRF-Chem simulated data over the domain 4.






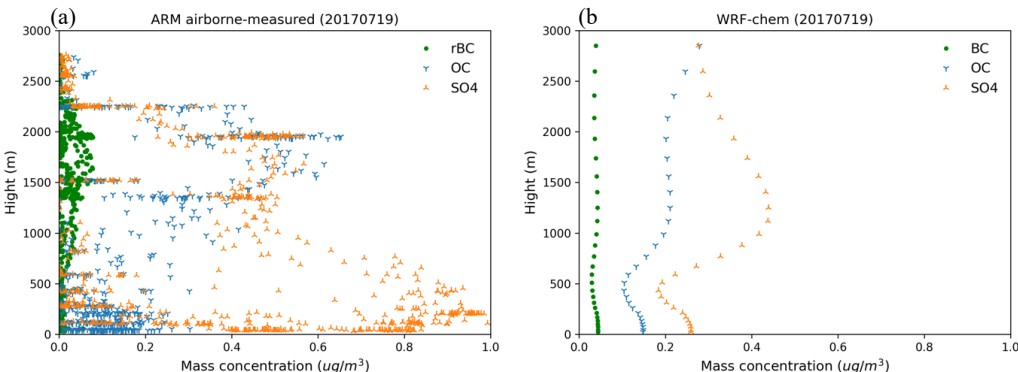

Figure 5. (a) ARM airborne-measured vertical profiles of SO$_4$, OC and refractory BC (rBC) mass
concentration (units: µg cm$^{-3}$) averaged over multiple flights on 19 July 2017. Noted that the
highly uncertain and noisy aerosol observations between 600 – 1000 m height due to cloud
contamination. (b) WRF-Chem simulated vertical profile of SO$_4$, OC and BC mass concentration
(units: µg cm$^{-3}$) averaged over the domain 4 during the flight time from 8:40 to 11:50 UTC.



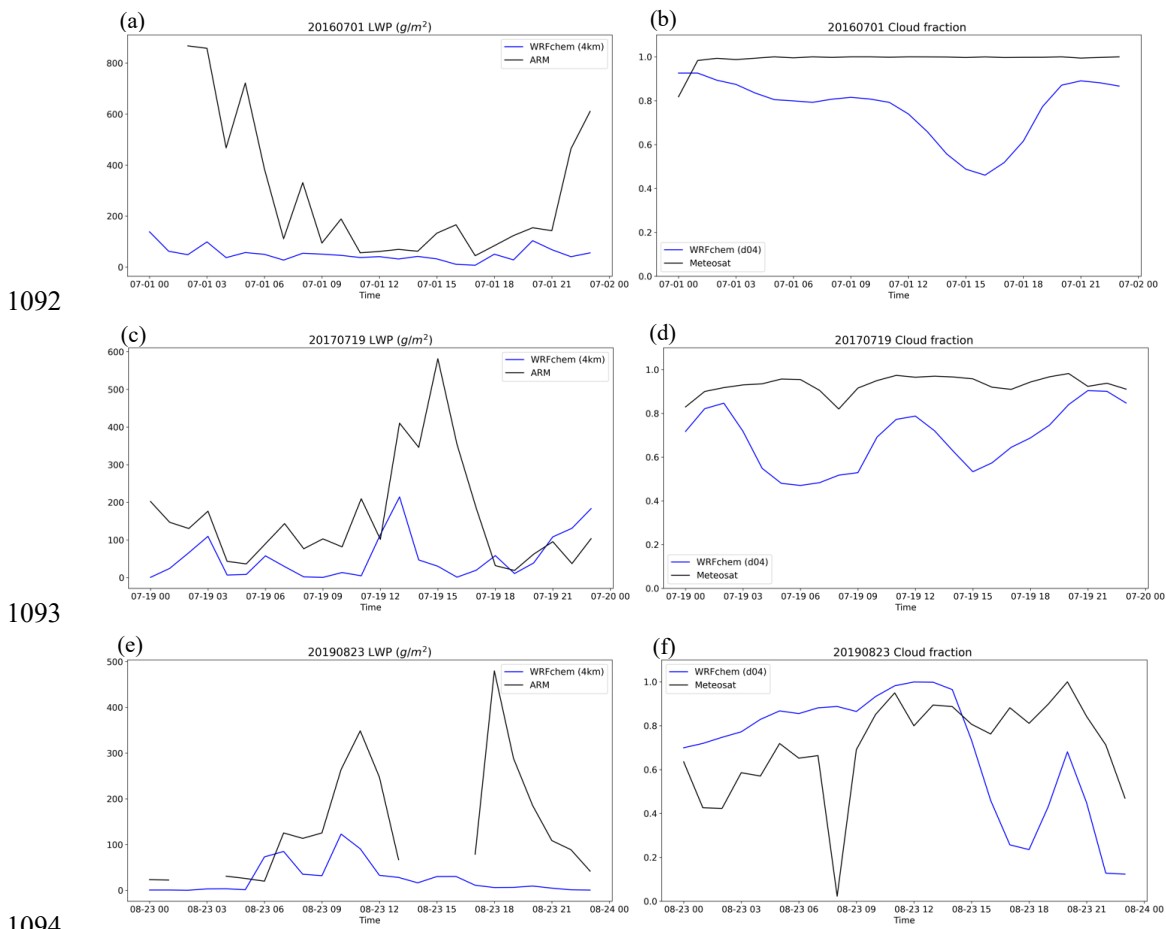

Figure 6. (a), (c), and (e) are the hourly time series of 4 km-averaged (4km) liquid water path (units: g m$^{-2}$) simulated from WRF-Chem (blue solid line) and observaed from ARM (black solid line) on 1 July 2016, 19 July 2017, and 23 August 2019, respectively. (b), (d), and (f) are the hourly time series of domain-averaged (d04) cloud fraction simulated from WRF-Chem (blue solid line) and observaed from Meteosat (black solid line) on 1 July 2016, 19 July 2017, and 23 August 2019, respectively. The 4 km-averaged data are averaged from the model simulated results over 20 × 20 grids centered on the Azores (approximately 4 km resolution).



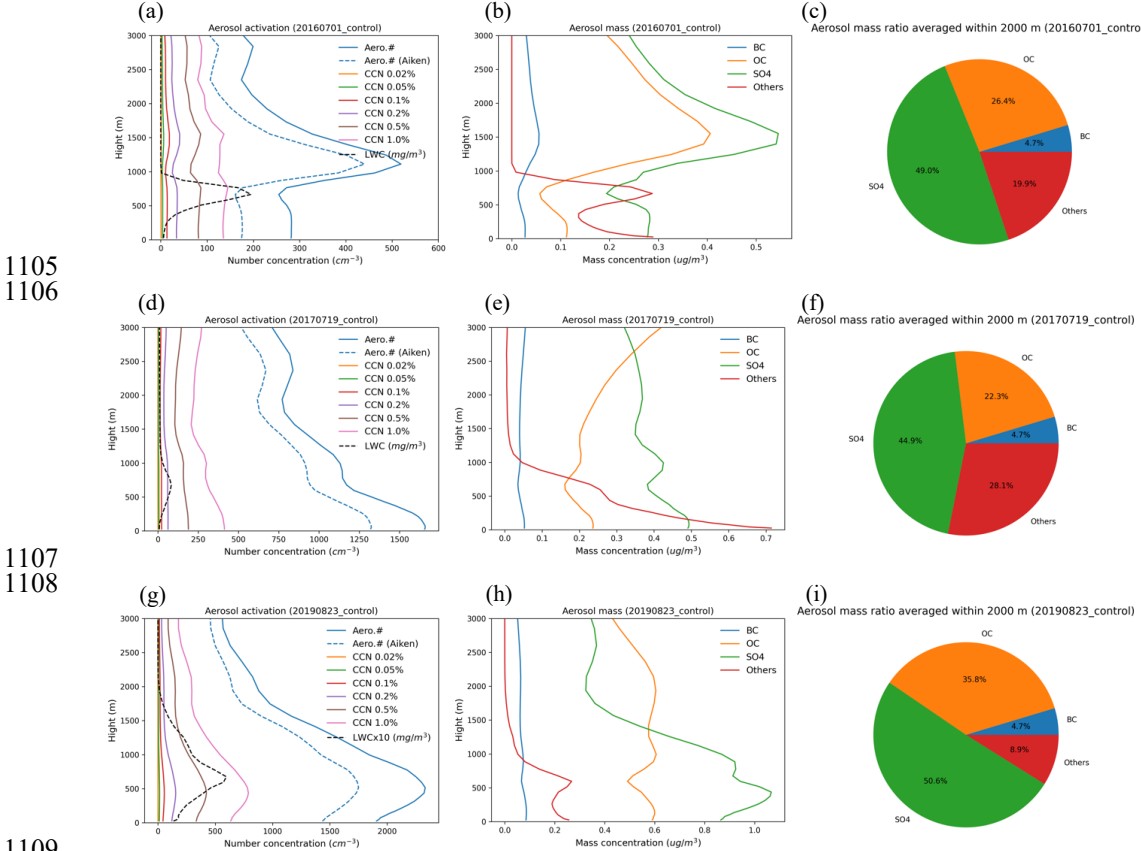

Figure 7. (a), (d), and (g) WRF-Chem vertical profiles of aerosol number concentration (Aiken mode and accumulation mode; units: cm$^{-3}$), aerosol number concentration (Aiken mode only; units: cm$^{-3}$), CCN number concentration under different supersaturations (units: cm$^{-3}$), and liquid water content (cloud and rain; units: mg m$^{-3}$) averaged over the domain 4 on 1 July 2016, 19 July 2017, and 23 August 2019, respectively, in the control runs. (b), (e), and (h) WRF-Chem vertical profiles of BC, OC, SO$_4$, and other species (like sea salts) (units: μg cm$^{-3}$) averaged over the domain 4 on 1 July 2016, 19 July 2017, and 23 August 2019, respectively, in the control runs. (c), (f), and (i) Pie chart of aerosol mass of different species averaged within 2000 m height on 1 July 2016, 19 July 2017, and 23 August 2019, respectively, in the control runs. Note that LWC quantity is adjusted to fit the scale of x-axis for each case.

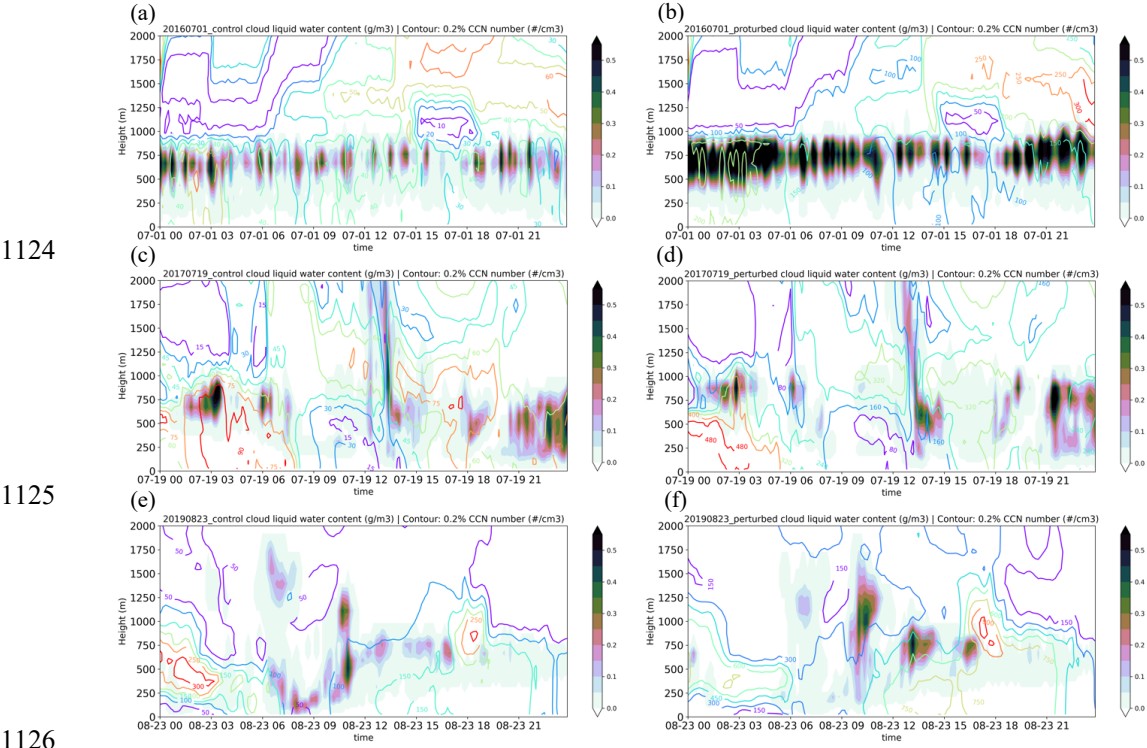

Figure 8. (a), (c), and (e) are the time series of 4 km-averaged cloud liquid water content profile (shade; units: g cm$^{-3}$) and CCN (0.2% supersaturation) number concentration profile (contour; units: # cm$^{-3}$) on 1 July 2016, 19 July 2017, and 23 August 2019, respectively, in the control runs. (b), (d), and (f) are the same as (a), (c), and (e), respectively, but in the perturbed runs. The data are averaged from the model simulated results over 20 × 20 grids centered on the Azores (approximately 4 km resolution).




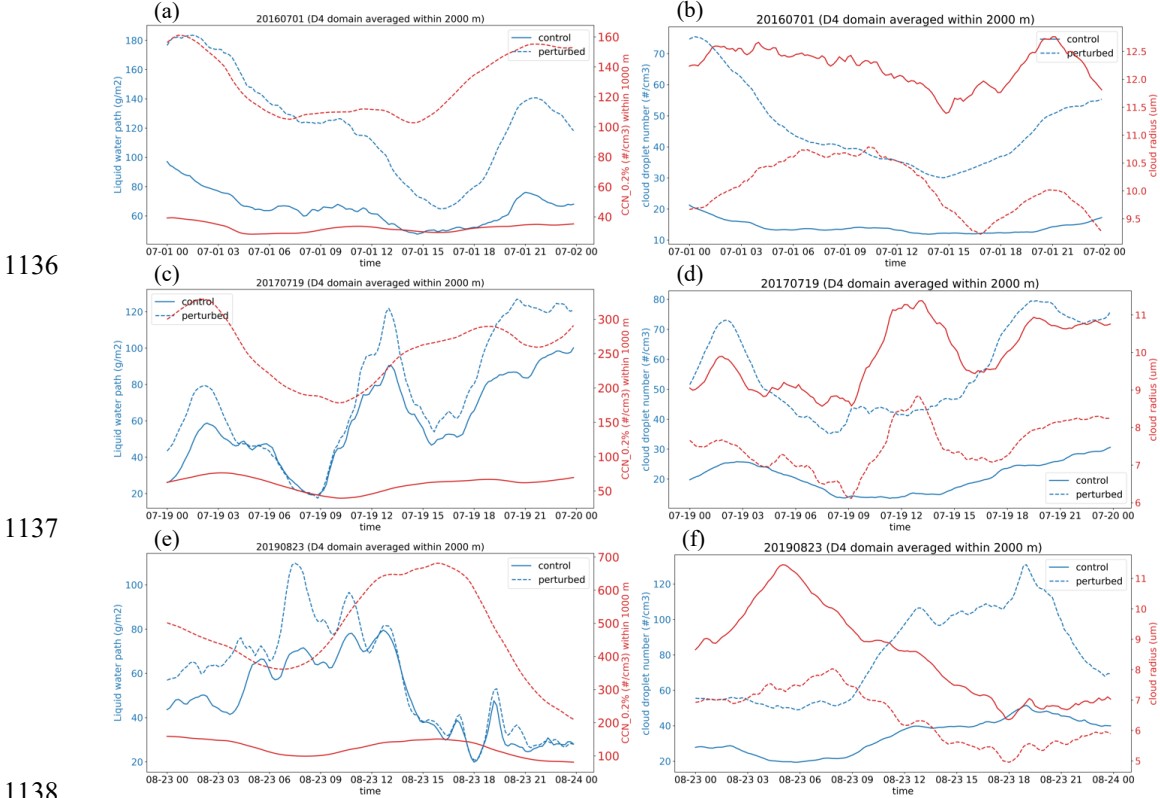

Figure 9. (a), (c), and (e) are the time series of domain-averaged liquid water path (blue lines; units: g m⁻²) and CCN number concentration under 0.2% supersaturation (red lines; units: # cm⁻²) for the control case (soild lines) and the perturbed case (dashed lines) on 1 July 2016, 19 July 2017, and 23 August 2019, respectively. (b), (d), and (f) are the time series of domain-averaged cloud droplet number (blue lines; units: # cm⁻³) and cloud radius (red lines; units: μm) for the control (soild lines) and perturbed (dashed lines) on 1 July 2016, 19 July 2017, and 23 August 2019, respectively. Only CCN data are averaged within 1000 m height over the domain 4, other variables are averaged within 2000 m height.







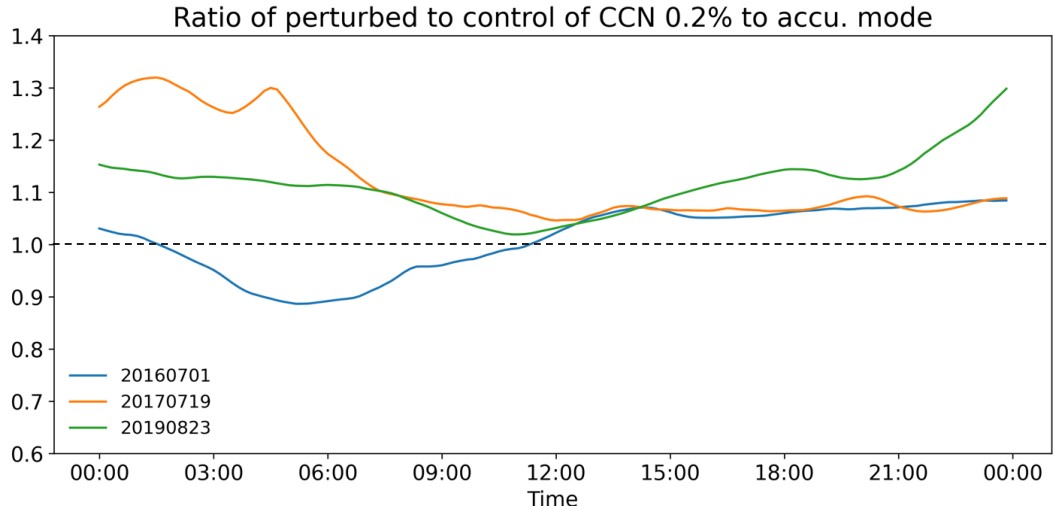

Figure 10. The time series of ratio of the precentage of activated CCN at 0.2% supersaturation to perturbed to the accumulation mode aerosols between perturbed and control runs. The back dashed line indicates the value of 1.0.

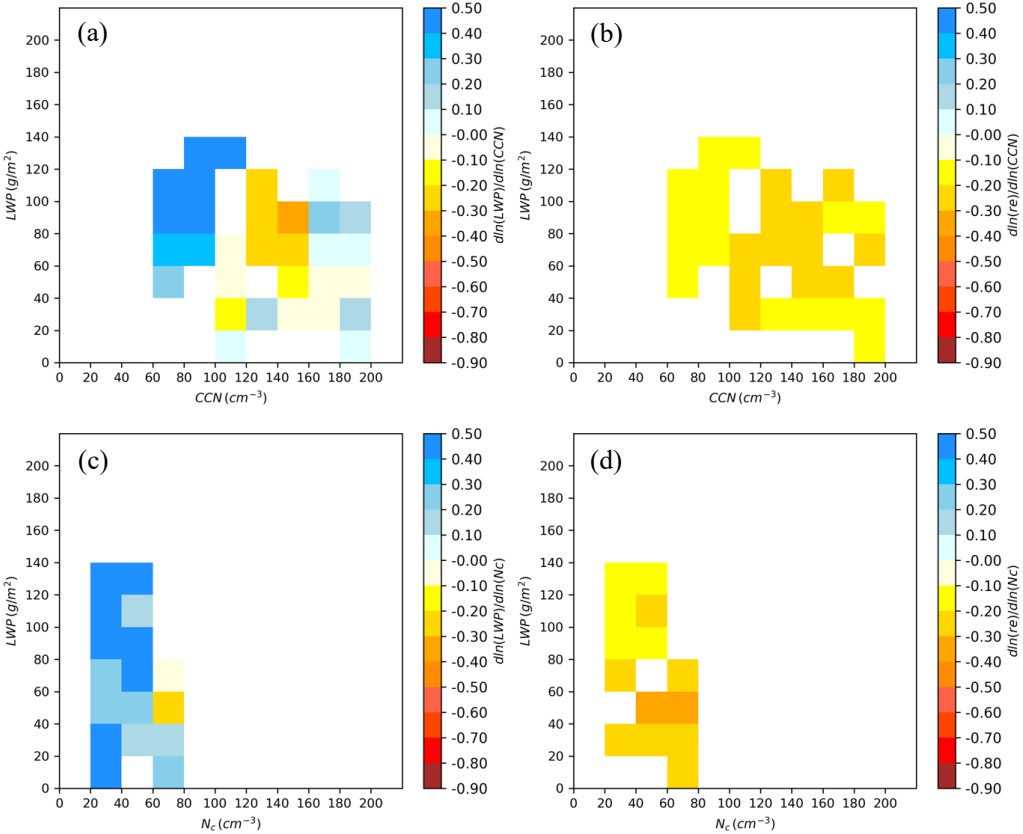


Figure 11. (a) and (b) are the mean liquid water path (LWP) and cloud radius (Re)
susceptibilities for different cloud condensation nuclei (CCN) and LWP bins for three study
cases, respectively. (c) and (d) are the same as (a) and (b), respectively, but for different cloud
droplet number (Nc) and LWP bins. The logarithmic slope between LWP and CCN, denoted as
($dln(LWP)/dln(CCN)$), is calculated at each output time (every 10 minutes) using data from 16
aggregate grid points (~25 km for each grid point) from the control run and 16 aggregated grid
points from the perturbed run.






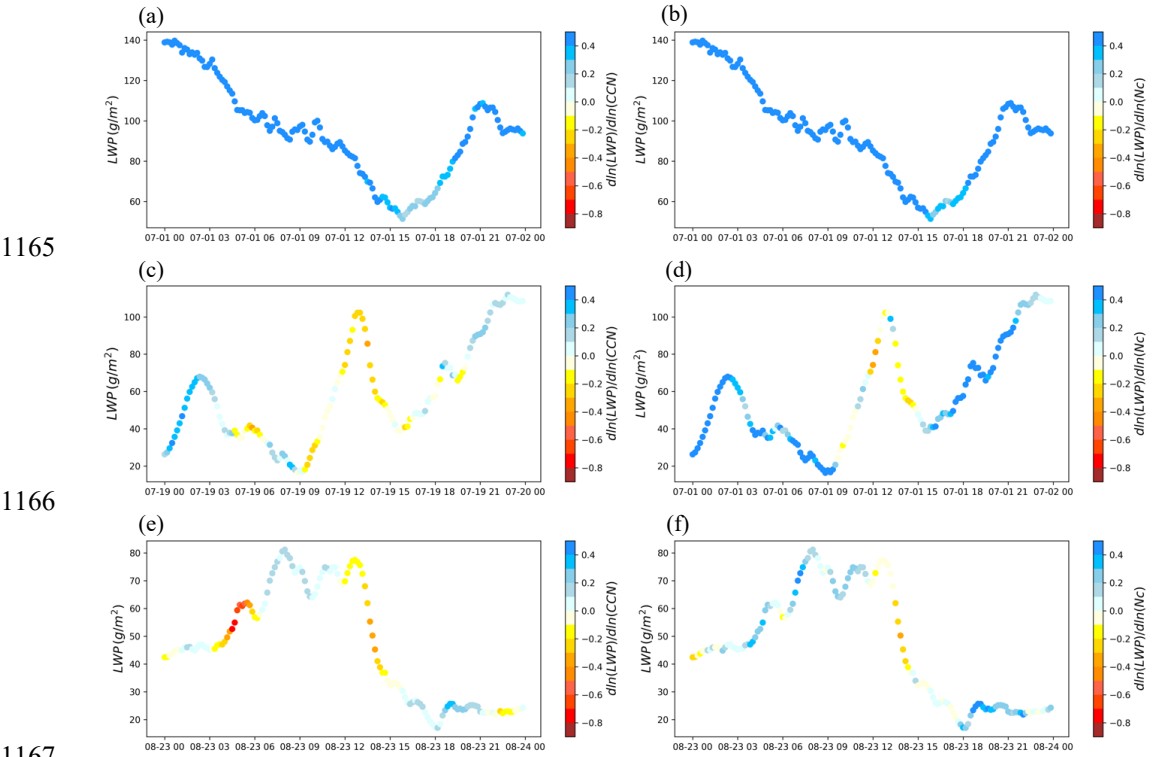

Figure 12. (a), (c) and (e) are the time variable of LWP susceptibility for different CCN
concentration, denoted as ($dln(LWP)/dln(CCN)$), on 1 July 2016, 19 July 2017, and 23 August
2019, respectively. (b), (d) and (f) are the time variable of LWP susceptibility for different Nc
concentration, denoted as ($dln(LWP)/dln(Nc)$), on 1 July 2016, 19 July 2017, and 23 August
2019, respectively. The logarithmic slope between LWP and CCN is calculated at each output
time (every 10 minutes) using data from 16 aggregate grid points (~25 km for each grid point)
from the control run and 16 aggregated grid points from the perturbed run.

1175



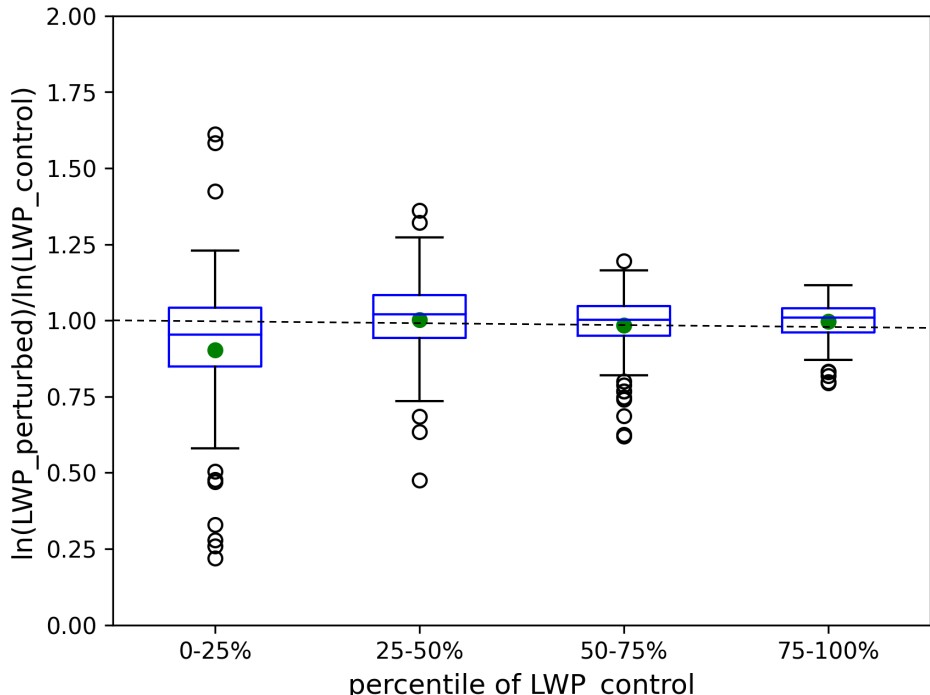

1176

Figure 13. The boxplot of the relative change in $ln(LWP)$ between the perturbed and control
cases across different LWP percentile rages in the control case during the negative susceptibility
for LWP shown in Fig. 12. The box extends from the first quartile to the third quartile of the
data, with a line at the median. The whiskers extend from the box to the farthest data point lying
within 1.5x the inter-quartile range from the box. Flier points are those past the end of the
whiskers. Green dots are the mean value, and the back dashed line indicates the value of 1.0.

1183