# Peer review of "Numerical Case Study of the Aerosol-Cloud-Interactions in Warm Boundary Layer Clouds over the Eastern North Atlantic with an Interactive Chemistry Module Hsiang-He Lee1, Xue Zheng1, Shaoyue Qiu1, and Yuan Wang2 1Atmospheric, Earth, and Energy Di"

_EGUsphere, 2024_

## Referee Comment (RC2)

Review of "Numerical Case Study of the Aerosol-Cloud-Interactions in Warm Boundary Layer Clouds over the Eastern North Atlantic with an Interactive Chemistry Module" by Lee et al. [Research Article, egusphere-2024-3199]

This study evaluated the simulations of three stratocumulus cloud cases in the ENA regions, each influenced by distinct weather regimes, using the WRF-Chem model. A key strength of this model is its ability to simulate aerosol-cloud interactions (ACIs) more realistically, thanks to its incorporation of aerosol chemical components and its consideration of aerosol spatiotemporal variations. The authors found that the model captured the liquid water path (LWP) and cloud fraction across the three cloud cases. They further investigated ACIs by conducting aerosol perturbation experiments, revealing a significantly positive LWP susceptibility in precipitating clouds due to precipitation suppression. They also identified some signals of negative LWP susceptibility, driven by aerosol drying effects in non-precipitating clouds, particularly at the cloud edges. Overall, the paper is well-organized and well-written, and the sensitivity experiments are thoughtfully designed. The findings regarding how LWP responds to aerosol perturbations under different weather regimes have important implications for improving stratocumulus simulations in the future. However, I have some concerns about the baseline simulation biases and their potential impact on the modeled ACIs. Addressing these issues would enhance the robustness of the study. If these concerns are resolved, I believe this paper will be suitable for publication in ACP.

**Major comments:**

1. The authors emphasize that the WRF-Chem model can potentially better represent cloud macrophysics due to its incorporation of aerosol chemical components and spatiotemporal variations. However, the model significantly underestimated LWP in all three cases, particularly the peak values, as illustrated in Figure 6. So, I am curious whether WRF-Chem can genuinely improve stratocumulus simulations through its refined representation of aerosol processes. Did the authors compare these results with those obtained using the standard WRF version? Additionally, the authors provided some potential explanations for the large LWP biases. For instance, in the 20170719 case, they suggested that these biases might stem from delayed moisture transfer from the outer domain or insufficient vertical resolution. Have higher vertical resolution simulations been tested to assess potential improvements? Also, including a precipitation evaluation in Figure 6 could help interpret the LWP and CF biases.

I think the accurate simulation of LWP in baseline simulations is the premise for the subsequent LWP susceptibility studies. If the simulated LWP in non-precipitating clouds is biased too low, the results might show a weaker negative LWP susceptibility or even no signal due to reduced cloud-top entrainment. Similarly, biases in LWP for precipitating clouds could impact the positive LWP susceptibility. If further improvements to LWP simulations are not feasible, the potential influence of these biases on the LWP susceptibility findings should be at least discussed.

2. Regarding LWP susceptibilities, I suggest that the authors expand their discussion by comparing the modeling results with observations, such as nearby ship tracks or observed LWP susceptibilities

at the ENA site. Such a comparison would help identify potential biases in the model's representation of cloud physics and enhance the broader implications of this study.

The authors investigated the variation of LWP susceptibility over time, finding positive susceptibilities during periods of no rain or light rain and negative susceptibilities during rain periods. However, previous studies (e.g., Hoffmann et al., 2024) typically observed positive susceptibilities in precipitating clouds and negative susceptibilities in non-precipitating clouds. Could the authors reconcile these seemingly contradictory findings?

3. The method used by the authors to aggregate simulation grids to calculate LWP susceptibility is not very clear. Including an illustration could help readers better understand this process. Also, it appears that the authors assumed a linear relationship between LWP and CCN in log space when calculating LWP susceptibility. However, this relationship might be non-linear or exhibit a reversed "V" shape, particularly when both precipitating and non-precipitating cases are included (e.g., Hoffmann et al., 2024). How did the authors account for the potential non-linear nature of this relationship when calculating LWP susceptibilities?

**Some technical suggestions:**

1. In many figures, the font size (including titles, axis labels, and tick labels) is too small to read. I suggest increasing the font size for better readability.
2. Please include Local Solar Time in the figures when discussing diurnal cycles to provide clearer context.
3. Please highlight raining periods in the relevant time series figures using colored boxes to make these periods easier to identify.

**Minor comments:**

L53: "Large-Eddy Simulation scale" to "large-eddy scale"

L55: I'd suggest reconsidering the phrase "accurately capturing the LWP and CF," as the large biases shown in Figure 6 do not fully support this claim.

L158: Could you clarify whether "cloud fraction" here refers to the low-level cloud fraction or the liquid cloud fraction? What is its precise definition?

L179: Is it in-cloud LWP or aggregated into grid-mean LWP? Please clarify.

L218: Please clarify domain sizes for each domain.

L222: What is the vertical resolution near the PBL top? Using a finer resolution here will help improve the representation of entrainment processes.

L250-253: Is there any chance to validate the calculated aerosol number concentration against observations?

L302: May also check the modeled CF.

L305: It seems like simulated clouds are overly scattered. Any reasons for that?

L310: For Figure 3, I'd suggest replacing absolute temperature with potential temperature that better depicts the static stability of the lower atmosphere. Also, please mark the inversion height for better readability. Please correct typos: "(b" and "(d" to "(b)" and "(d)".

L313: Which period for the biased inversion layer height?

L314: "and shows in Figs. S1a" to "as shown in Figs. S1a"

L319: Which period?

L330: I'd expect a stronger simulated inversion for the lower simulated PBLH. But why is the inversion layer weaker compared to observations?

L351-353: Is it because winds fields are better simulated in free troposphere than in the PBL?

L385: Please add local solar time in Figure 6.

L386: Please clarify what 4 km- and domain-average mean in the main text. Also, which domain is larger?

L414: Have you checked a shorter moisture input?

L419-421: Physically speaking, is it possibly due to stronger simulated shallow convection, which penetrates stratiform decks and break them up?

L427: Please mark the time period of two systems in the figure.

L433: The underestimation of LWP and CF, right?

L434: Do you mean a weaker cloud-top entrainment due to reduced cloud-top radiative cooling, leading to a shallower PBLH?

L457: Please mark the cloud top height in the figure.

L465: "which is the assumptions" to "specifically the assumption"

L480: "the most aerosols within" to "the most aerosols are within"

L487: Revised to high Aiken-mode aerosol number concentrations.

L517: "three" to "four"

L550-551: Does it precipitate during this period?

L559: Any reasons?

L569-571: Please provide an illustration to clarify this.

L614: "motioned" to "mentioned"

L625: It seems that you are referring to the 20160701 case. Please double check.

L657: I am curious why the LWP susceptibilities are not positive during rain events, given the precipitation suppression effect of aerosols, especially since aerosol perturbations are introduced at the start of the simulations. The explanation of negative susceptibilities due to aerosol-enhanced evaporation seems more plausible if aerosols were added immediately after the rainfall events.

L669: Is it possible the thin cloud deck rather than the cloud edges? Any evidence supporting your assumption?

L704: "struggle" to "struggles"

L707-708: If so, please justify why these cases are still appropriate as baseline simulations for understanding ACIs. (or see major comment #1).

L722-723: "not only promotes evaporation but also" to "promotes evaporation, thereby leading to"

L1097: "observaed" to "observed"

L1176: "percentile" to "Percentile"

Reference:

Hoffmann, F., Glassmeier, F., & Feingold, G. (2024). The impact of aerosol on cloud water: a heuristic perspective. *Atmospheric Chemistry and Physics*, *24*(23), 13403–13412. https://doi.org/10.5194/acp-24-13403-2024

---

## Author Response (AR1)

"Numerical Case Study of the Aerosol-Cloud-Interactions in Warm Boundary Layer Clouds over the Eastern North Atlantic with an Interactive Chemistry Module" by Lee et al. [Research Article, egusphere-2024-3199]

**Responses to the Comments of the Anonymous Referee #1**

We greatly appreciate the editing and feedback provided by this reviewer. In addition to the editing suggestions, which have been incorporated into the revised manuscript, we have outlined our point-by-point responses to the reviewer's questions and comments below (the reviewer's comments are indicated in italics).

***Comments:***

*Lines 263-265: There should be a more clear reason for your adjustment for sea salt emissions. Is this due to wind speed, observed emission data, etc?*

We revised the sentences to "Sea salt emissions are driven by surface wind speed. The simulated surface wind speed aligns closely with ERA-5 data; however, the sea salt concentration is only one-third of the value found in the MERRA-2 analysis. To improve the alignment with the sea salt aerosol concentration observed in the MERRA-2 reanalysis, we adjust the parameter factor for sea salt emissions to three times the original estimate (further comparison can be found in Section 3)." Lines 267-272 in the revised manuscript.

*Line 282: what is the cloud type?*

We clarify it is stratocumulus clouds in the manuscript.

*Lines 379-381: Is this part of the reasoning the authors chose to adjust sea salt concentration by three times?*

Yes. We also adjust the sentence to "Although the simulated surface wind speed matches well with ERA-5 (Fig. S7), the underestimation of sea salt concentrations may be attributed to limitations in the emission parameterization, which is overly reliant on surface wind speed (Gong, 2003)." Lines 391-394 in the revised manuscript.

*Line 392: is this a correct domain resolution complete with units?*

The resolution of domain 4 is 190 m, meaning that the aggregation of 20 grids results in a total distance of 3.8 km, which is approximately equivalent to a resolution of 4 km in the text.

*Line 472: below is a better word choice, revise for all instances.*

Modified "under" to "below".

*Line 519: Within a 1000 height layer? At 1000 m in altitude? What does this mean?*

We changed the sentence to "the CCN number concentration is averaged from the surface up to the height of 1000 m".

*Lines 641-644: These sentences seem to contradict one another. An increase in CCN has a large impact on LWP as shown by AIE. However, if the concentration exceeds 100 cm-3 (an increase) then there is a smaller sensitivity? Please revise.*

We thank the reviewer pointed it out. We modified the sentences to "However, when the mean CCN concentration exceeds 100 $cm^{-3}$, the relationship between LWP and CCN becomes more complex, with both positive and negative susceptibilities observed.  This suggests that the change in LWP is influenced by other factors, such as environmental conditions and cloud precipitation status.  (as shown in Fig. 11a)." Lines 676-679 in the revised manuscript.

*Line 672: NCCN?*

We changed it to CCN number.

*Figure 10: What specifically is this the ratio of? This sentence is awkward.*

The caption is changed to "The time series of the ratio of the number concentration of CCN at a supersaturation of 0.2% in the perturbed runs to that in the control runs, normalized by the corresponding accumulation mode aerosol concentration, defined as $\left. (CCN_{0.2\%}/Accu.\,aerosols)_{perturbed} \middle/ (CCN_{0.2\%}/Accu.\,aerosols)_{control} \right.$. The black dashed line indicates the value of unity."

*Figure 11: The cases are not labeled. There are no titles on the plots.*

The title is added to each plot. The LWP susceptibility plots include 6 simulations (both control and perturbed simulations of three cases).

*Figure 12: See previous comments for Figure 11.*

The title is added to each plot.

"Numerical Case Study of the Aerosol-Cloud-Interactions in Warm Boundary Layer Clouds over the Eastern North Atlantic with an Interactive Chemistry Module" by Lee et al. [Research Article, egusphere-2024-3199]

**Responses to the Comments of the Anonymous Referee #2**

We very much appreciate the constructive comments and suggestions from this reviewer. Our point-by-point responses to the reviewer's comments are as follows (the reviewer's comments are marked in Italic fort).

*Comments:*

*This study evaluated the simulations of three stratocumulus cloud cases in the ENA regions, each influenced by distinct weather regimes, using the WRF-Chem model. A key strength of this model is its ability to simulate aerosol-cloud interactions (ACIs) more realistically, thanks to its incorporation of aerosol chemical components and its consideration of aerosol spatiotemporal variations. The authors found that the model captured the liquid water path (LWP) and cloud fraction across the three cloud cases. They further investigated ACIs by conducting aerosol perturbation experiments, revealing a significantly positive LWP susceptibility in precipitating clouds due to precipitation suppression. They also identified some signals of negative LWP susceptibility, driven by aerosol drying effects in non-precipitating clouds, particularly at the cloud edges. Overall, the paper is well-organized and well-written, and the sensitivity experiments are thoughtfully designed. The findings regarding how LWP responds to aerosol perturbations under different weather regimes have important implications for improving stratocumulus simulations in the future. However, I have some concerns about the baseline simulation biases and their potential impact on the modeled ACIs. Addressing these issues would enhance the robustness of the study. If these concerns are resolved, I believe this paper will be suitable for publication in ACP.*

*Major comments:*

*1. The authors emphasize that the WRF-Chem model can potentially better represent cloud macrophysics due to its incorporation of aerosol chemical components and spatiotemporal variations. However, the model significantly underestimated LWP in all three cases, particularly the peak values, as illustrated in Figure 6. So, I am curious whether WRF-Chem can genuinely improve stratocumulus simulations through its refined representation of aerosol processes. Did the authors compare these results with those obtained using the standard WRF version? Additionally, the authors provided some potential explanations for the large LWP biases. For instance, in the 20170719 case, they suggested that these biases might stem from delayed moisture transfer from the outer domain or insufficient vertical resolution. Have higher vertical resolution simulations been tested to assess potential improvements? Also, including a precipitation evaluation in Figure 6 could help interpret the LWP and CF biases.*

*I think the accurate simulation of LWP in baseline simulations is the premise for the subsequent
LWP susceptibility studies. If the simulated LWP in non-precipitating clouds is biased too low,
the results might show a weaker negative LWP susceptibility or even no signal due to reduced
cloudtop entrainment. Similarly, biases in LWP for precipitating clouds could impact the positive
LWP susceptibility. If further improvements to LWP simulations are not feasible, the potential
influence of these biases on the LWP susceptibility findings should be at least discussed.*

We appreciate the reviewer's comments. In the original manuscript, we outlined three potential
reasons for the low biases in simulated LWP, as discussed in Section 3.1.3:

1. **Delayed moisture input from the boundary condition:** At the beginning of the project,
   we ran a 4-domain WRF-Chem simulation for the case of 20160701
   (20160701_4DChem) for only a few hours due to the limitation of computational
   resource. The simulated LWP in 20160701_4DChem is significantly higher than in other
   simulations. Figure A1a shows the simulated LWP in 20160701_4DChem at 00 UTC,
   which is much greater than the LWP even in 20160701_perturbed (see Fig. A1d).
   Additionally, the cloud fraction in 20160701_4DChem covers the entire domain 4,
   closely matching the values from Meteosat. In the 20160701_4DChem simulation,
   moisture transfer from D2 to D3 occurs every 10 seconds, whereas in the nested
   experimental design, it occurs every 5 minutes.

[Figure]

[Figure]

Figure A1. The simulated liquid water path (LWP; kg m$^{-2}$) over domain 4 on 1 July 2016 at 0 UTC from (a) 4-domain WRF-Chem (20160701_4DChem), (b) 2-domain nesting WRF, (c) 20160701_control, and (d) 20160701_perturbed.

2. **Vertical Resolution**: In another ongoing project, we increased the vertical layers to 99, which is to double vertical layers below 2km. However, the test with 99 levels only slightly improved the cloud cover and LWP. The estimated LWP susceptibility shows little to no change. The insignificant improvement indicates even higher resolution is needed to fully simulate the cloud processes near the sharp boundary layer inversion for these solid stratocumulus cloud layer.

3. **6th Order Horizontal Diffusion**: The 6th Order Horizontal Diffusion option used in this study (diff_6th_opt = 2) rapidly dissipates marine stratiform clouds, particularly at high spatial resolutions (Knievel et al., 2007). We tested using diff_6th_opt = 1, which resulted in a higher cloud amount; however, it also led to unreasonably high aerosol concentrations due to certain numerical issues. Consequently, we opted to use diff_6th_opt = 2 for this study.

We also mentioned that that the LWP retrieved by MWR at the ARM-ENA site experiences significant uncertainties during drizzling or precipitating conditions. This is primarily due to the scattering effects of large raindrops and raindrops accumulating on the instrument's radome, which can result in an overestimation of LWP.

In response to the reviewer's comment, Fig. A1b illustrates LWP simulated by a two-domain nesting WRF model, which shares a similar configuration with the control run but excludes chemistry. This simulation is based on a prescribed aerosol concentration of 200 cm$^{-3}$. The LWP in the 20160701_control run (Fig. A1c) is lower than that of the WRF simulation, while the LWP in the 20160701_perturbed run (Fig. A1d) aligns more closely with expected values. This indicates that aerosol activation is insufficient in 20160701_control. Conversely, the LWP and cloud droplet number in the 20160701_perturbed run are more consistent with observations and

the WRF simulation. This suggests that the perturbed run more accurately reflects reality, while the control run resembles a pristine environment for the case of 20160701.

Fig. A2 presents a comparison of LWP and CF among the control runs, WRF 2-domain nesting runs, and the observations. The WRF simulations were conducted with a prescribed aerosol number of 200 cm$^{-3}$. Except for the case of 20160701, the results for the other two cases show similar biases between WRF and WRF-Chem. The stratocumulus clouds during the 20160701 weather regime are particularly sensitive to aerosol changes. The perturbed simulation for this date aligns more closely with the observations (see Figs. 6a and 6b in the revised version). We acknowledge that the simulations have limitations in accurately reproducing cloud amounts, and we have documented the challenges encountered and provided a comprehensive discussion in the revised manuscript.

[Figure]

Figure A2. (a), (c), and (e) are the hourly time series of 4 km-averaged (4km) liquid water path (units: g m$^{-2}$) simulated from WRF-Chem (blue solid line), WRF (blue dashed line) and observed from ARM (black solid line) on 1 July 2016, 19 July 2017, and 23 August 2019, respectively. (b), (d), and (f) are the hourly time series of domain-averaged (d04) cloud fraction simulated from WRF-Chem (blue solid line), WRF (blue dashed) and observaed from Meteosat (black solid line) on 1 July 2016, 19 July 2017, and 23 August 2019, respectively. The 4 km-averaged data are averaged from the model simulated results over 20 × 20 grids centered on the Azores (approximately 4 km resolution).

In the revised manuscript, besides outlined three potential reasons for the low biases in simulated LWP, we added the following sentences in Lines 450-458:

"The underestimation of LWP and CF in model simulations leads to insufficient longwave cooling at the cloud top. This reduced cooling weakens cloud-top entrainment, resulting in a less pronounced boundary layer inversion and a shallower boundary layer height identified in the Section 3.1.1. This creates a negative feedback loop, where the initial inaccuracies in cloud properties affect boundary layer dynamics (Zheng et al., 2021). On the other hand, in the perturbed runs, the results show an adjustment in the amounts of LWP and CF, aligning more closely with the observations. This suggests that the CCN number is underestimated in the control runs (more discussion in Section 3.3). The model's response to aerosol changes highlights its capability for studying aerosol-cloud interactions."

2. *Regarding LWP susceptibilities, I suggest that the authors expand their discussion by comparing the modeling results with observations, such as nearby ship tracks or observed LWP susceptibilities at the ENA site. Such a comparison would help identify potential biases in the model's representation of cloud physics and enhance the broader implications of this study.*

This study focuses on developing the large-domain, near-LES resolution simulations driven by realistic conditions with interactive aerosol processes and initially assessing aerosol-cloud interactions using this proposed framework. The model framework shows potential for investigating aerosol-cloud interactions under various meteorological and aerosol conditions. It also enables the simulation of observational aerosol perturbations, such as ship tracks. Future work will aim to improve the identified model issues, such as the representation of transported aerosol size distributions, overly broken stratocumulus cloud layers with sharp boundary layer inversions and the comparison of the modeled cloud susceptibilities with observations through simulations of ship tracks and local aerosol perturbations.

In the revised manuscript, we added the following sentences in Section 4, Lines 801-805:

"Moreover, future research will focus on addressing the identified issues within the model, such as improving the representation of transported aerosol size distributions, resolving the overly fragmented stratocumulus cloud layers with sharp boundary layer inversions, and comparing the modeled cloud susceptibilities with observational data through simulations of ship tracks and local aerosol perturbations."

*The authors investigated the variation of LWP susceptibility over time, finding positive susceptibilities during periods of no rain or light rain and negative susceptibilities during rain periods. However, previous studies (e.g., Hoffmann et al., 2024) typically observed positive susceptibilities in precipitating clouds and negative susceptibilities in non-precipitating clouds. Could the authors reconcile these seemingly contradictory findings?*

We appreciate the reviewer's comments and the suggested reference. The study in Hoffmann et al. (2024) presents a heuristic model aimed at enhancing our understanding of how aerosols influence cloud water adjustments in stratocumulus clouds. By predicting the evolution of the

cloud water path (L) based on the cloud droplet number concentration (N), this model serves as a significant step forward in facilitating a process-level understanding of cloud water adjustments.

In our study, when we investigate the variation of LWP susceptibility over time, we observe that positive susceptibilities for different LWP and CCN (Nc) typically occur during periods of no rain or light rain. Such discrepancy is likely due the spatial heterogeneity of precipitating clouds shown in Fig. S10, and the non-linear relations between CCN and LWP as mentioned in comment #3. At 25-km scale, we are able to capture such spatial heterogeneity and retrieve a negative susceptibility due to the evaporation of thin clouds at cloud edge while at 100-km scale, the signal is dominated by the increase of LWP at cloud core.

As shown in Fig. S10, the slope representing the LWP susceptibility is based on 16 aggregated grid points, which reflect a different cloud behavior in global climate models with an approximate resolution of 1 degree. As a result, this slope is primarily influenced by the drying effect from thin, non-rain clouds mostly located at the edges of cloud cover.

To clarify this, in the revised manuscript, we added the following sentences in Section 3.4, Lines 709-721:

"In this study, the logarithmic slope between LWP and CCN in the LWP susceptibility calculation is based on a linear assumption. Hoffmann et al. (2024) introduced a heuristic model that represents a significant advancement in understanding the process-level adjustments of cloud water in stratocumulus clouds, suggesting that the relationship may resemble a reversed "V" shape. Figure 13 indicates that the decrease in LWP (negative susceptibilities) in the perturbed cases occurs only in low LWP clouds (thin and non-rain clouds). Conversely, LWP increases (positive susceptibilities) in thicker, precipitating clouds under the perturbed scenarios, which is consistent with the findings of Hoffmann et al. (2024). Since our study aggregates grids to a 25 km resolution, we are able to capture such spatial heterogeneity and retrieve a negative susceptibility due to the evaporation of thin clouds at cloud edge while at 100-km scale (domain-averaged shown in Figs. S11 and S12), the signal is dominated by the increase of LWP at cloud core. However, based on the current study cases, we may not have sufficient data samples to illustrate a relationship beyond the linear assumption."

3.      *The method used by the authors to aggregate simulation grids to calculate LWP susceptibility is not very clear. Including an illustration could help readers better understand this process. Also, it appears that the authors assumed a linear relationship between LWP and CCN in log space when calculating LWP susceptibility. However, this relationship might be non-linear or exhibit a reversed "V" shape, particularly when both precipitating and non-precipitating cases are included (e.g., Hoffmann et al., 2024). How did the authors account for the potential non-linear nature of this relationship when calculating LWP susceptibilities?*

We illustrate how we aggregated simulation grids in Fig. S10 to enhance the understanding of the susceptibility calculation. We acknowledge the reviewer's observation that the relationship may resemble a reversed "V" shape, as shown in Hoffmann et al. (2024). In Fig. 13, the boxplot indicates that the decrease in LWP in the perturbed cases occurs only in thin and no-rain clouds.

In contrast, LWP increases in thicker, precipitating clouds under the perturbed cases, which aligns with the findings of Hoffmann et al. (2024).

We revised manuscript and please see the reply to the major comment #2.

**Some technical suggestions:**

1. *In many figures, the font size (including titles, axis labels, and tick labels) is too small to read. I suggest increasing the font size for better readability.*
   Besides Figs 1, 2, 5, 11 and 13, the font size in other figures including figures in supplementary is adjusted to a bigger size.

2. *Please include Local Solar Time in the figures when discussing diurnal cycles to provide clearer context.*
   All time series figures in the manuscript are represented in local time, which is UTC – 1 hour.

3. *Please highlight raining periods in the relevant time series figures using colored boxes to make these periods easier to identify.*

   In Fig. 6, the red dots indicate when rainfall intensity is higher than 0.001 (0.01) mm hr$^{-1}$ in 4-km averaged area (domain 4), as well as in Fig. 12.

**Minor comments:**

*L53: "Large-Eddy Simulation scale" to "large-eddy scale"*

Modified.

*L55: I'd suggest reconsidering the phrase "accurately capturing the LWP and CF," as the large biases shown in Figure 6 do not fully support this claim.*

The sentence is modified to "The WRF-Chem simulations conducted at a near large-eddy scale offer valuable insights into the model's performance, especially regarding its high spatial resolution in capturing the mesoscale cloud features across various weather regimes." (Lines 53-55 in the revised manuscript.)

*L158: Could you clarify whether "cloud fraction" here refers to the low-level cloud fraction or the liquid cloud fraction? What is its precise definition?*

The cloud fraction from Meteosat is for all clouds detected by the cloud mask algorithm of SatCORPS. However, when we selected the case, we focused on warm marine boundary layer clouds defined by both ARM cloud radar and Meteosat retrievals. We modified the manuscript to "Specifically, this study adopts cloud fraction for all clouds as the observational reference over the ENA region" in Lines 156-157.

*L179: Is it in-cloud LWP or aggregated into grid-mean LWP? Please clarify.*

The ground-based observed LWP is retrieved from the brightness temperature measured by the microwave radiometer and the modeled LWP is in-cloud LWP. The modeled LWP is clarified in Section 3.1.1, Line 311 as" The modeled LWP is calculated in-cloud LWP only".

*L218: Please clarify domain sizes for each domain.*

We modified the manuscript to "There are $550 \times 530$ grids for D1, $451 \times 430$ grids for D2, $553 \times 532$ grids for D3, and $553 \times 532$ grids for D4. The domain size of domain 4 is about 1 degree which is similar to the spacing resolution of global climate models." in Section 2.2.2, Lines 219-221.

*L222: What is the vertical resolution near the PBL top? Using a finer resolution here will help improve the representation of entrainment processes.*

The vertical resolution near the PBL top is about 150 m. In this study, we increased the vertical layer from the WRF default setting 30 layers to 75 layers, and most increased layers within PBL. We also tried 99 vertical layers, and, indeed, a finer resolution can help improve the representation of entrainment processes. However, the model still underestimates the LWP for the study cases. Due to the limitation of computational resources, the 75 vertical layers are the best option for this study.

Please also see the reply to the major comment #1.

*L250-253: Is there any chance to validate the calculated aerosol number concentration against observations?*

We have tried our best to compare the modeled aerosol concentration to both the MERRA-2 reanalysis data and aircraft in situ observation. However, the most available data provides only aerosol mass concentration. Aerosol optical depth retrieved from satellite remote sensing can provide additional information, but it could be high bias in our study cases due to cloud cover.

 We added the following sentence to the revised manuscript, Lines 255-257:

" Aerosol optical depth retrieved from satellite remote sensing can offer valuable information for comparison; however, it may be subject to high bias in our study cases due to cloud cover."

*L302: May also check the modeled CF.*

We added the modeled CF to Fig. S1 in the revised supplementary.

*L305: It seems like simulated clouds are overly scattered. Any reasons for that?*

The color bar may have been misleading. We have adjusted the color bar for Figs. 2g-2l to better emphasize the thin clouds in these figures. Additionally, we included the perturbed runs to illustrate the changes in LWP compared to the control runs.

*L310: For Figure 3, I'd suggest replacing absolute temperature with potential temperature that better depicts the static stability of the lower atmosphere. Also, please mark the inversion height for better readability. Please correct typos: "(b" and "(d" to "(b)" and "(d)".*

We would like to keep the absolute temperature plots in the manuscript. We also added potential temperature plots (Fig. S2 in the revised version) in the supplementary. All typos are corrected.

*L313: Which period for the biased inversion layer height?*

The biased inversion layer height starts 23 Z local time on 30 June until the end of simulation. Figure 3b shows the simulated bias compared to Fig. 3a.

*L314: "and shows in Figs. S1a" to "as shown in Figs. S1a"*

Modified.

*L319: Which period?*

About the noon time. The sentence is modified to "However, compared to observations, the model does not catch the inversion at 1500 m height near the noon time and shows a warm bias in the model's simulated temperature (Figs. 3c and 3d)." in Lines 330-332.

*L330: I'd expect a stronger simulated inversion for the lower simulated PBLH. But why is the inversion layer weaker compared to observations?*

The weaker inversion layer could come from the parameterization schemes we chose in this study. The physical parameterizations used in the model (e.g., for turbulence or radiation) might not accurately represent the processes that strengthen the inversion layer. However, we think the most important is the feedback mechanisms. The underestimation of LWP and cloud fraction in the model simulations leads to insufficient longwave cooling at the cloud top. This insufficient cooling can weaken the cloud-top entrainment process, which is the mixing of air from the surrounding environment into the cloud. As a result of reduced cloud-top radiative cooling, the boundary layer inversion may be less pronounced, leading to a shallower PBLH. Thus, the weaker cooling reduces the stability of the boundary layer, allowing for a shallower depth. This creates a negative feedback loop, where the initial underestimation of cloud properties affects the boundary layer dynamics.

Please also see the reply to the major comment #1.

*L351-353: Is it because winds fields are better simulated in free troposphere than in the PBL?*

In the most numerical models, the wind fields are simulated better in free troposphere than in the PBL. In general, our simulation can capture the surface wind well (Fig. S7). However, there are still some spatiotemporal bias.

*L385: Please add local solar time in Figure 6.*

Modified.

*L386: Please clarify what 4 km- and domain-average mean in the main text. Also, which domain is larger?*

We added the domain size in Section 2.2.2. 4 km-average means 20x20 grids and domain-average means 553x532 grids. Domain-average covers larger area.

*L414: Have you checked a shorter moisture input?*

Yes, in a test run of 4-domain WRF-Chem (10-second timesteps for D2), the cloud structure is simulated better then 2-domain WRF-Chem with downscaling boundary condition (5 mins), but the computational cost is too high to complete the study.

Please see the reply to the major comment #1.

*L419-421: Physically speaking, is it possibly due to stronger simulated shallow convection, which penetrates stratiform decks and break them up?*

It is possible, especially for the case of 20160701. We conducted the same simulation settings for WRF to compare the results with WRF-Chem (see the reply to the comment #1). For the case of 20160701, the WRF simulation shows a better representation of cloud cover; however, the results do not improve in the other two cases. For those uniform, closed-cell clouds, the model with low aerosol concentration (20160701_control) may simulate stronger shallow convection, which penetrate stratiform decks and break them up.

We have outlined all potential challenges in the study, but the biases may arise from various factors.

*L427: Please mark the time period of two systems in the figure.*

We indicated the readers to see the high value of LWP observed from ARM (Fig. 6e).

*L433: The underestimation of LWP and CF, right?*

Yes. Corrected.

*L434: Do you mean a weaker cloud-top entrainment due to reduced cloud-top radiative cooling, leading to a shallower PBLH?*

Yes. The underestimation of LWP and cloud fraction in the model simulations leads to insufficient longwave cooling at the cloud top. This insufficient cooling can weaken the cloud-top entrainment process, which is the mixing of air from the surrounding environment into the cloud. As a result of reduced cloud-top radiative cooling, the boundary layer inversion may be less pronounced, leading to a shallower PBLH. Thus, the weaker cooling reduces the stability of the boundary layer, allowing for a shallower depth. This creates a negative feedback loop, where the initial underestimation of cloud properties affects the boundary layer dynamics.

Please see the reply to the major comment #1.

*L457: Please mark the cloud top height in the figure.*

The black dashed lines indicate the liquid water content which shows the cloud vertical distribution and cloud top height.

We changed the sentence to be more precise "The environment shown in the figure is characterized by its cleanliness, with a total aerosol number concentration below the cloud top (approximately 1000 m in height) measuring less than 300 $cm^{-3}$" in Lines 478-480.

*L465: "which is the assumptions" to "specifically the assumption"*

Based on the reviewer 1 suggestion, the sentence is changed to "This discrepancy arises from the assumptions made when constructing the aerosol initial and boundary conditions (80% for Aiken mode and 20% for accumulation mode $SO_4$)" in Lines 486-488.

*L480: "the most aerosols within" to "the most aerosols are within"*

Modified.

*L487: Revised to high Aiken-mode aerosol number concentrations.*

The sentence is changed to "Because of the high number concentration of simulated Aiken mode aerosols, overall, the activation rate is low."

*L517: "three" to "four"*

Changed.

*L550-551: Does it precipitate during this period?*

Yes. Please see Fig. S9c. We also indicated the precipitation time in Figs. 6 and 12.

*L559: Any reasons?*

Among the three cases, the case from 20190823 exhibits the highest aerosol concentration and CCN number, even in the control run (see Figs. 4 and 9). In our simulations, after the system passed away around noontime, more aerosols are transported to the area following the frontal system. As a result, more aerosols activated as CCN, leading to a more pronounced difference in CCN numbers and cloud droplet numbers between the control and perturbed runs.

We added this sentence in the manuscript "It is because aerosols are transported to the area following the frontal system (Fig. 4f) and then more aerosols activated as CCN" in Lines 592-594.

*L569-571: Please provide an illustration to clarify this.*

Please see Fig. S10.

*L614: "motioned" to "mentioned"*

Modified.

*L625: It seems that you are referring to the 20160701 case. Please double check.*

We corrected the typo. It is the 20160701 case.

*L657: I am curious why the LWP susceptibilities are not positive during rain events, given the precipitation suppression effect of aerosols, especially since aerosol perturbations are introduced at the start of the simulations. The explanation of negative susceptibilities due to aerosol-enhanced evaporation seems more plausible if aerosols were added immediately after the rainfall events.*

Please see the replies for major comment #2 and #3.

*L669: Is it possible the thin cloud deck rather than the cloud edges? Any evidence supporting your assumption?*

The thin cloud deck is correct as well, especially we show the relative change in ln(LWP) between perturbed and control cases based on the percentile of LWP (Fig. 13).

We modified the sentences to "The results indicate a decrease in LWP in the perturbed cases compared to the control cases for pixels with the lowest LWP percentile range (0-25%). We assume that this range corresponds to thin clouds at the edges of the cloud cover." In Lines 702-704.

*L704: "struggle" to "struggles"*

Modified.

*L707-708: If so, please justify why these cases are still appropriate as baseline simulations for understanding ACIs. (or see major comment #1).*

Our findings remain valuable for understanding ACIs across different weather regimes. They emphasize the susceptibility of clouds to aerosol variations in both closed cells (as demonstrated in the case of 20160710) and open cells (shown in the cases of 20170719 and 20190823). For example, in the 20160701 case, the cloud fraction in the perturbed run aligns more closely with observations, indicating a deficiency in aerosol activation within the model. The control run for 20160701 represents a more pristine environment, while the perturbed run is more reflective of real conditions.

In the other two cases, the effects of cloud drying are evident in the open cell clouds, further illustrating the ACI processes explored in this study. Additionally, our research shows that in-cloud collision–coalescence processes effectively remove aerosols, particularly due to the wider range of cloud droplet sizes in a clean environment (i.e., the control runs). This aligns with findings from Terai et al. (2014), which studied five pockets of open cells using aircraft data from the VOCALS Regional Experiment (VOCALS-REx). In their study, clouds break up easily if there is a lack of aerosols, i.e., a pristine environment, which stem from different reasons. Even in the control run of 20160701, our study shows the broken open cell clouds in a pristine environment is hard to develop a closed cloud.

In the revised manuscript, we added the following sentences in Section 3.3, Lines 536-545:

"Echoing the insufficient longwave cooling at the cloud top due to the underestimation of LWP and CF in model simulations discussed in Section 3.1.3, Terai et al. (2014) also observed weak cloud-top entrainment in their study of five pockets of open cells, using aircraft data from the VOCALS Regional Experiment (VOCALS-REx).  Their research indicated that clouds tend to break up easily in the absence of aerosols, characteristic of a pristine environment.  Consistent with their findings, our study demonstrates that in-cloud collision–coalescence processes effectively remove aerosols, particularly because of the wider range of cloud droplet sizes present in a clean environment (i.e., the control runs) (Table 2).  Even in the control run for 20160701, our results indicate that broken open cell clouds in a pristine environment struggle to develop into closed clouds."

*L722-723: "not only promotes evaporation but also" to "promotes evaporation, thereby leading to"*

Modified.

*L1097: "observaed" to "observed"*

Modified.

*L1176: "percentile" to "Percentile"*

Modified.

---

## Author Response (AR2)

"Numerical Case Study of the Aerosol-Cloud-Interactions in Warm Boundary Layer Clouds over the Eastern North Atlantic with an Interactive Chemistry Module" by Lee et al. [Research Article, egusphere-2024-3199]

**Responses to the Comments of the Anonymous Referee #1**

We appreciate the editing provided by this reviewer. The editing suggestions have been incorporated into the revised manuscript, and we have also outlined our point-by-point responses to the reviewer's comments below (the reviewer's comments are indicated in *italics*).

***Comments:***
*This manuscript is overall well written but still needs some minor edits which are enclosed in the PDF.*
*Line 590:… is characterized as clean,…*
Modified.

*Line 618: remove "the"*
Modified.

*Line 619: "cloud layer" what? Do you mean to say within the cloud layer?*
We modified the sentence to "In the simulation of 20170719_control, most aerosols are concentrated within a height of 1000 m, which corresponds to the cloud layer height"

*Lines 733-734: which in turn activates more aerosols as CCN.*
Thans for suggestions and the sentence is modified.

*Line 911: case studies*
Modified.

*Figures 3 and 4: remove minus sign.*
Modified.

*Figure 5: Be sure to use the correct time format for this journal.*
It is followed the time format requested for ACP.

*Figure 6: 4 km*
Changed in the figure caption.

*Figure 12: dashed*
Thank you for catching the typo. Modified.

*Be sure to remove the minus sign for western longitude in your supplemental figures.*
All figure captions in the supplementary are updated.

"Numerical Case Study of the Aerosol-Cloud-Interactions in Warm Boundary Layer Clouds over the Eastern North Atlantic with an Interactive Chemistry Module" by Lee et al. [Research Article, egusphere-2024-3199]

**Responses to the Comments of the Anonymous Referee #2**

We appreciate the constructive comments and suggestions from this reviewer. Our point-by-point responses to the reviewer's comments are as follows (the reviewer's comments are marked in *Italic* fort).

***Comments:***

*I appreciate the authors' efforts in addressing my comments. Most of them have been thoroughly resolved. I have only one remaining concern regarding the calculation of LWP susceptibility to CCN concentrations. In this study, the susceptibility is quantified through the logarithmic slope of aggregated grid points from both the control and perturbed simulations. However, such calculated slopes might be statistically insignificant (e.g., likely so in Figure S10c upon visual inspection). To enhance the robustness of the findings, I recommend that the authors include statistical significance information for each calculated susceptibility in Figure 12 (and Figure 11 if applicable). For example, they could use solid dots to indicate cases where $p <= 0.01$ and circles for $p > 0.01$. This might help reconcile the discrepancy with the results from Hoffmann et al. (2024). In other words, positive LWP susceptibilities observed during periods of no rain or light rain might not be significant.*

We appreciate the reviewer's suggestion and have updated our manuscript accordingly. The p-value for Figure S10c is $8.1 \times 10^{-9}$, confirming its strong statistical significance. The data points from the perturbated run have systematically lower LWP than these in the control run.

In response to the reviewer's comments, we have incorporated p-value calculations into Figure 11 and Figure 12. In Figure 12, black circles indicate statistically significant results ($p \leq 0.01$), with more than 70% of the results meeting this criterion. Similarly, Figure 11 highlights statistically significant results with black edges. Our findings support the conclusion that when CCN concentrations are low, LWP is highly sensitive to variations in CCN number, with higher CCN concentrations leading to increased LWP. However, when the mean CCN concentration is relatively high, LWP becomes less sensitive to changes in CCN, resulting in smaller LWP susceptibilities, which include both positive and negative values (most with $p > 0.01$). The negative values are primarily caused by evaporation of small cloud droplets at non-rain grid points near the cloud edges.

We have updated the figures and revised the relevant text to include our response to the reviewer's comments, as reflected in Lines 678-679.

***Editing suggestions:***
*Line 432: "which is to double" to "which doubles"*

Modified.

*Line 1246: "rages" to "ranges"*

Thank you for catching the typo. Modified.